METHODS AND RESOURCES

# A comparison of anatomic and cellular transcriptome structures across 40 human brain diseases

Yashar Zeighami[1,2]*, Trygve E. Bakken[3], Thomas Nickl-Jockschat[4], Zeru Peterson[4], Anil G. Jegga[5,6], Jeremy A. Miller[3], Jay Schulkin[7], Alan C. Evans[2], Ed S. Lein[3], Michael Hawrylycz[3,8]*

**1** Douglas Research Centre, Department of Psychiatry, McGill University, Montreal, Canada, **2** Montreal Neurological Institute, McGill University, Montreal, Canada, **3** Allen Institute for Brain Science, Seattle, Washington, United States of America, **4** Department of Psychiatry, Neuroscience and Pharmacology, Iowa Neuroscience Institute, University of Iowa, Iowa City, Iowa, United States of America, **5** Division of Biomedical Informatics, Cincinnati Children's Hospital Medical Center, Cincinnati, Ohio, United States of America, **6** Department of Pediatrics, College of Medicine, University of Cincinnati, Cincinnati, Ohio, United States of America, **7** Department of Obstetrics and Gynecology, University of Washington, Seattle, Washington, United States of America, **8** University of Washington, Department of Genome Sciences, Seattle, Washington, United States of America

* yashar.zeighami@mcgill.ca (YZ); mikeh@alleninstitute.org (MH)

**Data Availability Statement:** All data used in this manuscript are publicly available. The gene disease association data can be downloaded from https://www.disgenet.org/. The large-scale anatomic

## Abstract

Genes associated with risk for brain disease exhibit characteristic expression patterns that reflect both anatomical and cell type relationships. Brain-wide transcriptomic patterns of disease risk genes provide a molecular-based signature, based on differential co-expression, that is often unique to that disease. Brain diseases can be compared and aggregated based on the similarity of their signatures which often associates diseases from diverse phenotypic classes. Analysis of 40 common human brain diseases identifies 5 major transcriptional patterns, representing tumor-related, neurodegenerative, psychiatric and substance abuse, and 2 mixed groups of diseases affecting basal ganglia and hypothalamus. Further, for diseases with enriched expression in cortex, single-nucleus data in the middle temporal gyrus (MTG) exhibits a cell type expression gradient separating neurodegenerative, psychiatric, and substance abuse diseases, with unique excitatory cell type expression differentiating psychiatric diseases. Through mapping of homologous cell types between mouse and human, most disease risk genes are found to act in common cell types, while having species-specific expression in those types and preserving similar phenotypic classification within species. These results describe structural and cellular transcriptomic relationships of disease risk genes in the adult brain and provide a molecular-based strategy for classifying and comparing diseases, potentially identifying novel disease relationships.

## Introduction

Brain diseases are increasingly recognized as major causes of death and disability worldwide [1–3]. These diverse and multifactorial diseases may be largely grouped into cerebrovascular,

transcriptional patterns can be downloaded from http://human.brain-map.org/ and cell type data is available at http://celltypes.brain-map.org/. The script (Jupyter notebook) and the data files for producing the figures are provided at https://doi.org/10.5281/zenodo.7709525.

**Funding:** This work was in part supported by funding from the Canada First Research Excellence Fund, awarded to McGill University for the Healthy Brains, Healthy Lives (HBHL) initiative New Recruit Start-Up Supplements Program, as well as Réseau de Bio-Imagerie du Québec (RBIQ /QBIN). MH was also supported by R01MH123220 (PI) 08/01/ 2020-07/31/2022 (NIH): A Community Framework for Data-driven Brain Transcriptomic Cell Type Definition, Ontology, and Nomenclature grant. "The funders had no role in study design, data collection and analysis, decision to publish, or preparation of the manuscript."

**Competing interests:** The authors have declared that no competing interests exist.

**Abbreviations:** ADG, Anatomic Disease Group; AHBA, Allen Human Brain Atlas; ALM, anterior lateral motor; BICAN, Brain Initiative Cell Atlas Network; BICCN, Brain Initiative Cell Census Network; CGS, central glial substance; CN, cerebellar nuclei; DS, differential stability; ECT, electroconvulsive therapy; EWCE, expression-weighted cell type enrichment; FDR, false discovery rate; FL, frontal lobe; GBD, Global Burden of Disease; GDA, gene–disease association; GP, globus pallidus; GR, gracile nucleus; IHME, Institute for Health Metrics; MS, multiple sclerosis; MTG, middle temporal gyrus; OCD, obsessive-compulsive disorder; OMIM, Online Mendelian Inheritance in Man.

neurodegenerative, movement related, psychiatric disorders, developmental and congenital disorders, substance abuse disorders, brain tumors, and a set of other brain-related diseases (Institute for Health Metrics (IHME), healthdata.org). The economic impact of brain diseases also varies substantially, as reflected in the comprehensive and annually updated Global Burden of Disease Study [4] (**Fig A in S1 Text**). The etiology of brain-related diseases and their genetics is complex and widely studied [5–7]. However, phenotypic classification of brain diseases is challenging and does not uniquely partition characteristics of genetic risk, disease manifestation, and treatment. Except for mendelian diseases arising from single-gene mutations, most brain disorders present as a complex interplay between genetics and environment through interaction of the brain transcriptome and its regulatory network. Genetic analysis of brain disease, through profiling of tissues, cells, and more recently at the resolution of single nuclei [8] provides means for population scale sampling to disentangle basic molecular relationships [9,10].

Characterizing the neuroanatomy of major transcriptomic relationships for brain diseases and its relationship to cell type provides a novel means of disease comparison and classification. The premise of the present study is the hypothesis that spatial and temporal co-expression of disease genes is indicative of a potential interaction between these genes [11,12] and that disease aggregation based on these patterns is informative. Studying brain samples from donor populations exhibiting coherent transcriptomic and anatomic relationships of disease-related genes, both in neurotypical and diseased brains and at multiple scales, promises important insight in developing further approaches to study the pathophysiology of brain disorders, particularly as brain-wide cellular data becomes increasingly available. Large-scale transcriptome profiling of the human brain has already produced useful resources for exploring the genetics of neurotypical and disease states [13–16] and in describing the larger scale relationship of brain diseases and the neuroanatomy of transcriptomic patterning [13,17].

Transcriptomic relationships at a mesoscale, intermediate between the larger brain structures (e.g., cortex, hypothalamus) and those at cellular resolution, provide a framework and starting point for classifying broad disease associations in comparison with common phenotypic grouping. Starting with the Allen Human Brain Atlas (human.brain-map.org) [13,14], we investigated anatomic patterning and differential expression of the transcriptional patterns in the adult neurotypical brain of genes for 40 brain-related disorders across 104 structures from cortex, hippocampus, amygdala, basal ganglia, epithalamus, thalamus, ventral thalamus, hypothalamus, mesencephalon, cerebellum, pons, pontine nuclei, myelencephalon, ventricles, and white matter. Using single-nucleus data from the human middle temporal gyrus (MTG), we further characterize a subset of 24 diseases with primary expression in cortex by comparing expression of cell types from a taxonomy of 45 inhibitory, 24 excitatory, 6 non-neuronal types, and with special attention to psychiatric diseases. This multiresolution approach combining tissue-based and single-nucleus data connects mesoscale anatomic analysis with cell types of the cortex and is a recognized approach for extracting information from tissue-based sampling [18,19]. Finally, juxtaposing these results with single-cell data in mouse [15,20] allows identification of potential important human-specific cell type differences as well as insight into the overlapping mechanisms in animal models of brain disorders.

## Brain disorders and associated genes

The diseases selected are representative of 7 phenotypic classes from the Global Burden of Disease Study (referred to as **GBD** classes in this study). The important group of cerebrovascular diseases was excluded due to limitations of representative endothelial and pericyte cell types and related blood cells in data sources. To identify gene–disease associations (GDAs), we used

the DisGeNET database (www.disgenet.org) [21–23], a platform aggregated from multiple sources including curated repositories, GWAS catalogs, animal models, and the scientific literature. From an initial survey of the Online Mendelian Inheritance in Man (OMIM) (www.omim.org) repository, we previously identified 549 potential brain-related diseases [13] that are now intersected with the DisGeNET repository. We required reported GDAs to be present in at least 1 confirmed curated source (see https://www.disgenet.org/dbinfo) and with a minimum of 10 genes per disease. For each disease, the main variant of the disease was selected with rare familial and genetic forms not included. This conservative selection resulted in 40 major brain disorders with 1,646 unique associated genes. **S1 Table** contains definitions, gene sets, and metadata identifying each disease (**Methods**).

Gene sets associated with brain disease vary widely in size and the proportion of shared genes, and diseases can be associated by phenotypic similarity based on clinical manifestations [24,25]. The gene set sizes in this study range widely from frontotemporal lobar degeneration (g = 11) to schizophrenia (g = 733) and distribute widely across GBD classes as (number, % unique to GBD class) psychiatric (1,107, 0.723), neurodegenerative (257, 0.513), substance abuse (212, 0.320), brain tumors (168, 0.667), developmental disorders (139, 0.676), movement related (136, 0.272), and other brain related (123, 0.414) (**Fig B in S1 Text**). The large gene set intersection (g = 132) between psychiatric and substance abuse GBD classes, with 62% of substance abuse genes also associated with psychiatric disorders, reflects the well-established comorbidity of these diseases [26]. Movement disorders are also commonly found in neurodegenerative diseases [27], with neurodegenerative sharing 30% (g = 41) of movement-related genes, while GBD tumor based and developmental share the least with other classes (2.5% and 2.6%, respectively). Clustering the 40 diseases and disorders based on relative pairwise gene set intersection (Jaccard) shows moderate agreement with GBD phenotypic groupings (**Fig B in S1 Text**), with the highest percentage of shared genes among psychiatric disorders 7.64% ($p = 1.55 \times 10^{-4}$), followed by substance abuse 6.33% ($p = 2.82 \times 10^{-4}$), and brain tumors 5.43% ($p = 8.35 \times 10^{-3}$). (Significance is likelihood of observed percentage corrected for GBD class size.) Functional enrichment analysis (https://toppgene.cchmc.org) of genes unique to each GBD describes major biological processes and pathways of these groups (**Fig C in S1 Text and S2 Table**).

## Neuroanatomy and transcriptomic profiles of brain diseases

Expression profiles from the Allen Human Brain Atlas (AHBA, https://human.brain-map.org) from 6 neurotypical donor brains are used to summarize major neuroanatomical relationships of genes associated with the 40 diseases. Using an ontology of 104 structures (**S3 Table**) from cortex (CTX, 8 substructures), hippocampus (HIP, 7), amygdala (AMG, 6), basal ganglia (BG, 12), epithalamus (ET, 3), thalamus (TH, 12), hypothalamus (HY, 16), mesencephalon (MES, 11), cerebellum and cerebellar nuclei (CB, 4), pons and pontine nuclei (P, 10), myelencephalon (MY, 12), ventricles (V, 1), and white matter (WM, 2), we obtained a mean transcriptomic disease profile by averaging expression for genes associated with each of 40 diseases across the 104 structures and z-score normalizing (**Fig 1 and S4 Table**). Performing hierarchical clustering with Ward linkage using Pearson correlation (**Methods**) presents brain-wide transcriptomic associations in 5 primary Anatomic Disease Groups (**ADG 1–ADG 5**) interpretable with respect to GBD classification (**Fig 1A**, left color bar) as tumor related (**ADG 1**), neurodegenerative (**ADG 2**), psychiatric, substance abuse, and movement disorders (**ADG 3**), a group without developmental, psychiatric, or tumor diseases associated with hypothalamic function (**ADG 4**), and a group of diseases related to basal ganglia (**ADG 5**).

The anatomic representation of transcriptomic patterning within each ADG group is described as **ADG 1:** thalamus, brain stem, ventricle wall, white matter; **ADG 2:** cortico-

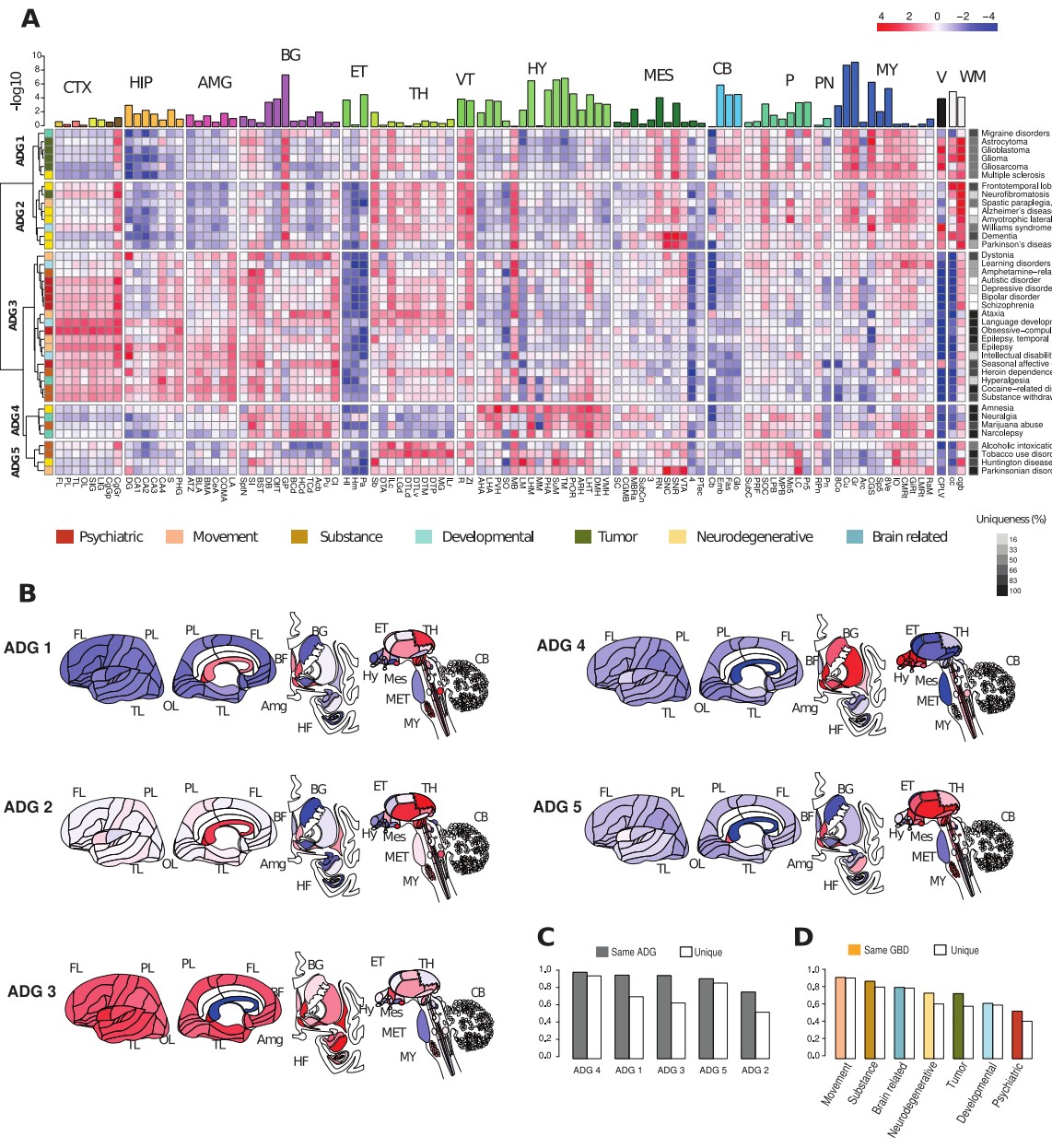

**Fig 1. Transcriptome patterning of major brain diseases.** (A) Mean gene expression profiles for genes associated with 40 major brain diseases and disorders profiled over 104 anatomic structures (**S3 Table**) from 15 major regions cortex (CTX), hippocampus (HIP), amygdala (AMG), basal ganglia (BG), epithalamus (ET), thalamus (TH), ventral thalamus (VT), hypothalamus (HY), mesencephalon (MES), cerebellum (CB), pons (P), pontine nuclei (PN), myelencephalon (MY), ventricles (V), white matter (WM). Hierarchical clustering based on z-score mean profile yields 5 primary anatomic disease groups **ADG 1–ADG 5**. Row annotation (left bar) shows phenotypic GBD membership with color codes. Column bar annotation is 5 group ANOVA for ADG expression variability at a fixed structure. Row annotation (right bar): number of genes associated with disease (log scale). (B) Brain graphic illustrating anatomic patterning of classes ADG 1–5. (C) Reproducibility of ADG profiles. (Solid) Frequency that an ADG disease transcriptomic signature is most closely correlated with a signature from the same ADG in other subjects. (Open) Frequency that exact disease is identified in other subjects. (D) Similar analysis for diseases by phenotypic GBD groups. (Solid) Same GBD class, (Open) exact disease agreement. Underlying data for Fig 1 can be found in S1 and S2 Tables, and the data from S1 Data HBA disease files. Raw data available at http://human.brain-map.org/. Code available as a notebook at https://github.com/yasharz/human-brain-disease-transcriptomics. ADG, Anatomic Disease Group; GBD, Global Burden of Disease.

thalamic, brain stem, white matter; **ADG 3:** (telencephalon) cortex, thalamus, hippocampus, amygdala, basal ganglia; **ADG 4:** basal ganglia, hypothalamus, brain stem; and **ADG 5:** thalamus, hypothalamus, brain stem. **Fig 1** illustrates the complex anatomic structure of disease gene expression and remarkably, the division and structure of **ADG** groups is largely preserved (67%) upon removing genes common between pairs of diseases (**Fig D in S1 Text**) showing that distinct co-expressing genes drive the major ADG groups. The clustering also remains stable subsampling the diseases having very large gene sets (**Fig E in S1 Text**). ADG transcriptome signatures are also consistent across subjects as individual brain holdout analysis (**Figs F and G in S1 Text, Methods**) finds that both the correlation of expression across structures and differential relationships between ADG groups at a fixed structure are preserved within the subjects.

Disease gene burden can vary significantly (from high burden to risk factor), and the strength of evidence supporting each gene varies where some are convergently supported by multiple large cohort studies, whereas others may have conflicting data. To account for these effects, we used the literature-based GDA weights provided by the DisGeNET dataset through a GDA score (**Methods**). Although there may be variability in accuracy of gene-disease weights, the result of the weighting analysis (**Fig H in S1 Text**) corroborates the disease associations of **Fig 1** with 85% agreement. Furthermore, the diseases presented have very different temporal genetic signatures and this may confound associations. We observe, however, that even genes that likely act mostly in development to cause pathology may continue to contribute to disease state in adulthood, and neurodevelopmental disorders have symptoms that are persistent across life span. While our analysis does not account for temporal dynamics, examination of the BrainSpan (https://www.brainspan.org) data using donors from 60 days old to 39 years (**Fig I in S1 Text**) highlights the expected temporal patterning and onset of expression, with clustering retaining many associations found in the adult.

The complex anatomic organization of gene expression reflected in **Fig 1** associates diseases with common phenotypic classification by the GBD study, but with important divergences (**Fig 1A**, left sidebar) that are supported by the literature. **ADG 1,** driven by co-expression in the diencephalon, myelencephalon, and white matter, comprises tumor-based diseases with the association of migraine disorders and multiple sclerosis (MS). The concurrence of MS and brain tumors has been widely described [28–30], and MS patients have decreased overall cancer risk, but an increased risk for brain tumors [31], a hypothesis being that remyelinating processes coincide with a decline of the CNS immune function. Patients with brain tumors also experience an increased risk of having a prior migraine diagnosis [32]. **ADG 2** comprises most of the neurodegenerative diseases, with the association of Williams syndrome and hereditary spastic paraplegia, and early aging, dementia, autoimmunity, and chronic inflammation are characteristics of diseases associated with oxidative stress [33]. Amyotrophic lateral sclerosis has been associated with Alzheimer disease ([34] as well as with frontotemporal dementia [35]. In addition to strong substantia nigra (SNC) expression in dementia and Parkinson's disease, this group has stronger expression in cortex and hypothalamus mammillary bodies, where abnormalities have been observed in neurodegeneration [36]. The common association of all psychiatric diseases, and most movement, and substance disorders in **ADG 3** is driven by strong telencephalic patterning. Psychiatric manifestations after occurrence of epilepsy have often been noted yet are not completely understood [35,37]. Seizures are known to be extremely effective modulators of psychiatric symptoms, and electroconvulsive therapy (ECT) still is used today as one of the most effective antidepressant and antipsychotic treatments. **ADG 4** comprises diseases from mixed phenotypic classes, with a consistent hypothalamic signature (**Fig 1C**), and where amnesia and narcolepsy may be associated with hypothalamic lesions [38,39], and narcolepsy with excess marijuana use [40,41]. Finally, **ADG 5** is dominated

by diseases affecting the basal ganglia Parkinsonian signs of bradykinesia in Huntington's disease have been found to typically manifest over time [42].

To understand the variability of expression across ADG groups, we apply ANOVA for mean differences in expression across at each structure (BH corrected $p$-values, top annotation, **Fig 1**). Particularly striking in **Fig 1A** is the white matter signature common to tumor and neurodegenerative diseases (**ADG 1–2**), effectively absent in psychiatric disorders and diseases of addiction (**ADG 3–4**) [43], and substantial enrichment of telencephalic expression (CTX, HIP, AMG, and BG) in **ADG 3** [44,45]. The most significant transcriptomic variation in disease genes across the adult brain occurs across the diverse nuclei of lower brain structures: in the hypothalamus (e.g., cuneate nucleus (Cu, $p < 3.35 \times 10^{-8}$), tuberomammillary nucleus (TM, $p < 1.3 \times 10^{-6}$), supramammillary nucleus (SuM, $p < 2.07 \times 10^{-6}$), in the myelencephalon (gracile nucleus (GR, $p < 1.48 \times 10^{-8}$)), central glial substance (CGS, $p < 3.86 \times 10^{-6}$), in the basal ganglia (globus pallidus (GP, $p < 5.01 \times 10^{-7}$)), and cerebellar nuclei (CN, and white matter, in particular, corpus callosum (cc, $p < 5.01 \times 10^{-7}$)). The distinction between **ADG 1** and **ADG 2** is more subtle with variation in cortex (frontal lobe, FL, $p < 2.82 \times 10^{-3}$), epithalamus (lateral habenular nucleus, (HI, $p < 2.85 \times 5$)), and mesencephalon (pretectal region, (PTec, $p < 6.24 \times 10^{-5}$)) and will be further examined using module-based analysis (**Fig 2**). For details see **Fig J in S1 Text**.

While the expression of disease genes may vary considerably in a population [46,47], the anatomic expression signature of each disease in an individual brain is typically closely correlated with a disease in the same ADG group in other brains (**Fig 1C**) (**ADG 1–5**: 96.7%, 77.0%, 96.1%, 100.0%, and 92.5%), and often identifies the exact disease in other subjects (**Figs 1C and 1G in S1 Text, Methods**). The ability to identify a disease from its expression signature provides a characterization of that disease by neuroanatomy. Surprisingly, the expression signature associated with the **ADG 3** group diseases ataxia, language development disorders, temporal lobe epilepsy, obsessive compulsive disorder, and cocaine-related disorder most closely correlates with these same diseases in each of the subjects. Similarly, in **ADG 4 and 5,** genes associated with Parkinsonian disorders, Huntington's disease, amnesia, narcolepsy, neuralgia, and tobacco use disorder exhibit unique profiles across subjects due to consistent, differentiated expression in the basal ganglia, hypothalamus, and myelencephalon. Conversely, the mesoscale transcriptomic profile of **ADG 2** Alzheimer's disease and amyotrophic lateral sclerosis, and **ADG 3** bipolar disorder, autistic disorder, and schizophrenia are less unique to those diseases, suggesting potential cellular, anatomical, and phenotypic overlaps between them and other disorders in the same ADG groups. Phenotypically, GBD movement disorders and substance abuse have the most consistent anatomic signatures (94.0%, 89.5%) (**Fig 1D**), while psychiatric and developmental diseases the least (64.0%, 55.0%).

## Canonical expression patterns of brain diseases

The neuroanatomy of transcription patterns for disease risk genes can be further characterized by directly identifying differential expression relationships and reproducible patterns that are conserved in the adult. By mapping disease genes to these canonical expression patterns [13], we describe the co-expressing patterns of the brain disorders and the major constituent cell types. Differential stability (DS), introduced in [13], is quantified as the mean Pearson correlation $\rho$ of expression between pairs of specimens over a fixed set of anatomic regions and measures the fraction of preserved differential relationships between anatomic regions for a set of subjects. For example, the gene *GRIA2* with remarkably high DS ($\rho = 0.918$), (**Fig 2A**) is implicated in bipolar disorder [48], schizophrenia [49], and substance withdrawal syndrome [50] and has a highly reproducible brain-wide expression profile across AHBA subjects with highest expression in hippocampus and amygdala.

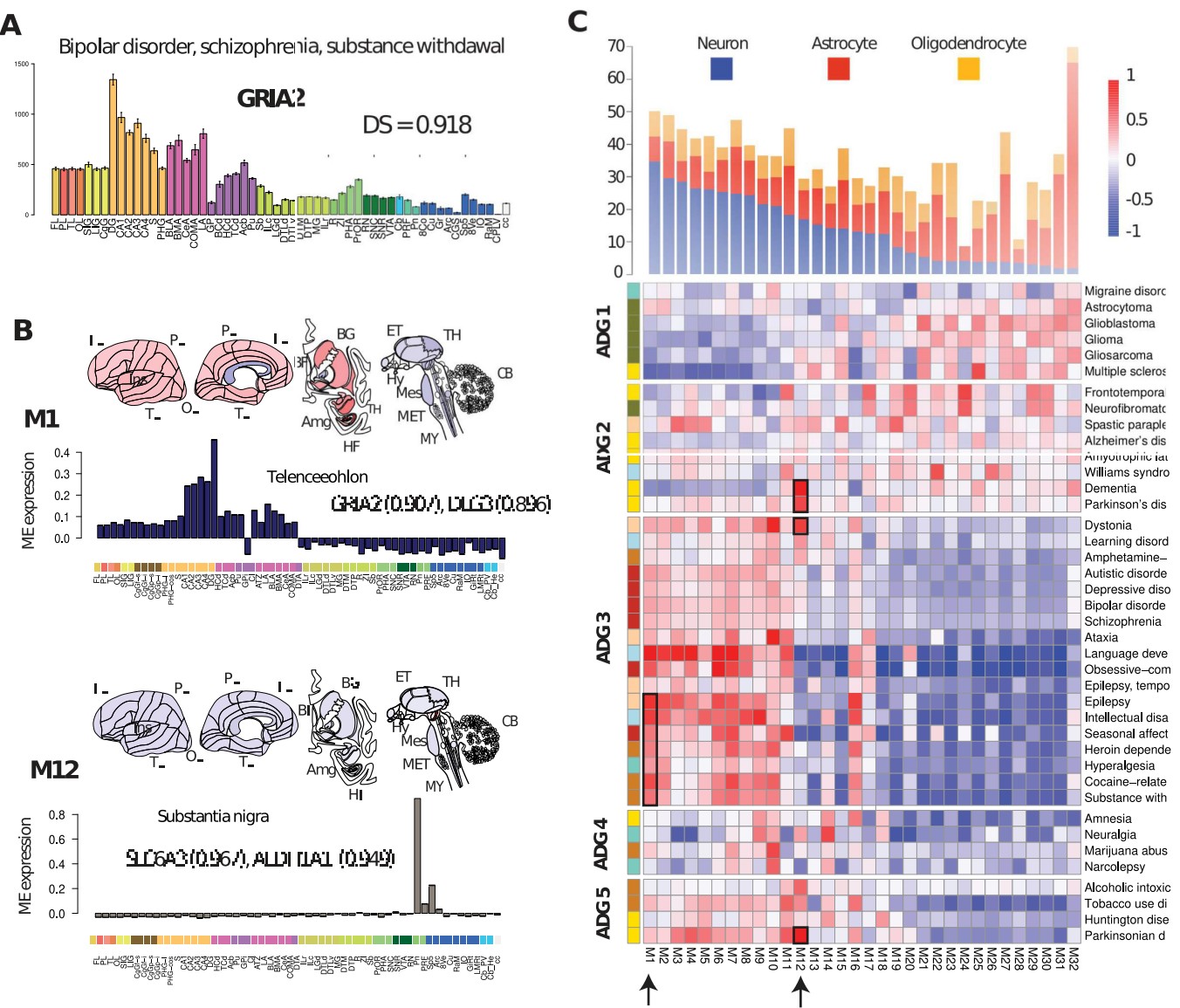

**Fig 2. Reproducible transcription patterns in human brain diseases.** (A) Expression profile for gene GRIA2 with error bars shown over 56 structures (S3 Table, human.brain-map.org). DS measures reproducible expression patterns from the AHBA [13]. (B) Canonical eigengenes for **M1** telencephalic (language development, epilepsy) and **M12** substantia nigra (Parkinson's disease, dementia), with module correlation for representative genes. (C) Map of canonical expression modules **M1**-**M32** mapping diseases to anatomic patterns. Disease genes are correlated with each module independently and normalized (**Methods**), disease ordering is the same as in **Fig 1**. Modules **M1**-**M32** are ordered based on their neuronal, astrocyte, oligodendrocyte cell type content derived in [13]. Arrows and boxes indicate diseases overrepresented in **M1** and **M12**. Other disease representative modules are described in **Fig M in S1 Text**. Underlying data for Fig 2 can be found in S1 and S5 Tables and the data from S1 Data disease module file. Raw data available at http://human.brain-map.org/. The information for canonical expression modules are available as S6 Table at https://www.nature.com/articles/nn.4171. Code available as a notebook at https://github.com/yasharz/human-brain-disease-transcriptomics. AHBA, Allen Human Brain Atlas; DS, differential stability.

Although disease genes show a marginally significant ($p < 0.031$) difference in their expression levels compared with non-disease associated genes (**panel A of Fig K in S1 Text**), disease genes have a significantly higher percentage of differentially stable genes, particularly for substance abuse, (mean DS = 0.702, $p < 4.7 \times 10^{-21}$), psychiatric (DS = 0.675, $p < 3.17 \times 10^{-17}$), and movement disorders (DS = 0.635, $p < 1.21 \times 10^{-17}$) (**panel B of Fig K in S1 Text**). DS prioritizes neuronal cell types with strong structural markers and less the non-neuronal broad

non-regional expression common in glial cells (**Fig L in S1 Text**). High DS disease genes are also substantially enriched for cell type processes, e.g., anterograde trans-synaptic signaling, (low DS $1.8 \times 10^{-4}$, high $2.9 \times 10^{-12}$), and presynaptic membrane (low DS 0.049, high $4.62 \times 10^{-7}$), indicating high DS selects for cell type specificity. Notably, the stability of genes in **ADG 3** (median 0.625, $p < 4.1 \times 10^{-70}$), **ADG 4** (0.642, $1.24 \times 10^{-6}$), and **ADG 5** (0.644, $2.95 \times 10^{-14}$) are markedly higher than **ADG 1** ($0.592 \times 10^{-6}$, 1.02) and **ADG 2** (0.582, $7.70 \times 10^{-7}$) indicating a higher percentage of neuronal cell types and structural markers in these groups. **Panel C of Fig K in S1 Text** shows the distribution of DS genes for each disease, confirming that diseases with higher DS are those with more anatomic structural markers.

A previous characterization of the reproducible gene co-expression patterns [13] in the Allen Human Brain Atlas using the top half of DS genes (DS > 0.5284, g = 8,674) identified 32 primary transcriptional patterns, or modules, each represented by a characteristic expression pattern (i.e., eigengene) across brain structures and ordered by cell type content. **Fig 2B** illustrates the membership of certain disease risk genes to modules for 2 representative modules **M1** and **M12**. Module **M1** has strong telencephalic expression in the hippocampus, in particular, dentate gyrus, and representative genes include *GRIA2* (correlation to eigengene, ρ = 0.907) and *DLG3* (ρ = 0.896).

Alterations in glutamatergic neurotransmission have known associations with psychiatric and neurodevelopmental disorders and mutations in *GRIA2* have been related with these disorders [46–48]. **M12** is a unique neuronal marker of substantia nigra *pars compacta*, *pars reticulata*, and ventral tegmental area and provides a clearer connection of dystonia, Parkinson's disease, and dementia for these comorbidities (**Fig 2C**). Both the dopamine transporter gene *SLC6A3* (ρ = 0.967), a candidate risk gene for dopamine or other toxins in the dopamine neurons [51,52] and aldehyde dehydrogenase-1 (*ALDH1A1*, ρ = 0.949), whose polymorphisms are implicated in alcohol use disorders, map to module **M12** (ρ = 0.949) [53]. Brain-wide association of expression module profiles may potentially be used to implicate genes without previous association to a given disease, particularly when that profile is highly conserved between donors.

A set of disease risk genes can be mapped to the canonical modules, by finding the closest correlated module eigengene for each gene, thereby providing the distribution of expression patterns associated with the disease (**S5 Table**). **Fig 2C** shows the normalized mean correlation of the 40 disease-associated gene sets with the module **M1-M32** eigengenes ordered by **ADG** as in **Fig 1** (**Methods**). The basic cell class composition of neuronal, oligodendrocyte, astrocyte of AHBA tissue samples was determined from earlier single-cell studies [13] and the modules **M1-M32** are ordered by decreasing proportion of neuron-enriched cells. Interestingly, **Fig 2C** clarifies the distinction between ADG groups of **Fig 1**, shows major cell type content, and illustrates the primary anatomic co-expression patterns of brain diseases.

Primarily tumor-based **ADG 1** maps to modules **M21-M32** having enriched glial content ($p < 2.413 \times 10^{-15}$), while **ADG 3** psychiatric and substance abuse-related diseases map to neuronal enriched patterned modules **M1-M10** ($p < 2.2 \times 10^{-16}$). Importantly, the neurodegenerative disorders of **ADG 2** including Alzheimer's, Parkinson's, ALS, and frontotemporal lobe degeneration show more uniform distribution across the modules, and now importantly separate this group from **ADG 1** ($p < 1.55 \times 10^{-15}$). **ADG 4 and 5** are both enriched in specific anatomic markers, e.g., **M10** (striatum), narcolepsy, marijuana, **M14** (hypothalamus), neuralgia, amnesia, **M11** (thalamus), Parkinsonian and tobacco use disorders, **M12** (substantia nigra), Parkinsonian, and alcoholic intoxication, yet have lower expression in neuronal modules **M1-12** than **ADG 3** (1-sided, $p < 3.84 \times 10^{-13}$). The distribution of **Fig 2C** validates the clustering of **Fig 1**, clarifies the distinction between ADGs and provides a classification of diseases through common transcriptional patterns and major constituent cell types (**Figs N and O in S1 Text**).

## Disease genes and cell types of middle temporal gyrus

A primary telencephalic expression pattern is common to diseases of **ADG 3,** and while meso-scale systems level analysis describes brain-wide anatomic relationships, it is limited in its ability to implicate specific cell types in diseases [12,54]. To examine these diseases more finely, we now restrict to those 24 diseases having higher than median cortical expression in the brain-wide analysis shown in **Figs 1 and 2**, essentially the entirety of **ADG 3** and several neurodegenerative diseases from **ADG 2.** We used human single nucleus (snRNA-seq) data from 8 donor brains (15,928 nuclei) from the MTG [15] where 75 transcriptomic distinct cell types were previously identified, including 45 inhibitory neuron types and 24 excitatory types as well as 6 non-neuronal cell types. A set of 142 marker genes are used to differentially distinguish the MTG cell types in [15]. These genes form a highly differentially stable group (DS = 0.734, $p < 8.66 \times 10^{-7}$), indicating strong cell type specificity, with 30 among the disease genes, several uniquely associated with a disease (**S6 Table**).

We measure the tendency for disease gene co-expression to enrich in a specific cell type, using the Tau-score ($\tau$) defined in [55] (**Methods**). For a gene $g$, $0 \leq \tau(g) \leq 1$ measures the tendency for expression to range from uniform across cell types to concentrated in a specific cell type. Averaging $\tau$ over sets of genes representing a given disease, we obtain a measure of cell type specificity of each disease within MTG (**panel C of Fig P in S1 Text**). Expression level differences between brain and non-brain disease genes while present ($p = 0.005$), are not as substantial as the significant difference in $\tau$ specificity between these groups $p < 2.2 \times 10^{-16}$ (**panels A and B of Fig P in S1 Text**) confirming specialized cell type involvement in genes associated with brain diseases. Pooling to the 7 GBD categories (**Fig 3B**), the genes from psychiatric ($p < 2.52 \times 10^{-74}$), movement ($p < 1.71 \times 10^{-11}$), and substance abuse disorders ($p < 3.58 \times 10^{-11}$) show the highest cell type specificity, while tumors, developmental disorders, and neurodegenerative diseases less.

**Fig 3A** presents the clustering of mean expression profiles across the 24 cortical brain diseases. Diseases are clustered by cell type specific expression and with annotations showing primary subclass level types (Inhibitory: *Lamp5*, *Pvalb*, *Sst*, *Sst Chodl*, *Vip*; Excitatory: *IT*, *NP*, *ET*, *CT*, *L6b*; and 5 non-neuronal types.) Cell type analysis in **Fig 3** identifies 4 primary Cell Type Groups (**CTG 1–4)** for these cortical diseases. Here, **CTG 1**, representing several movement and substance abuse disorders, is characterized by a strong enrichment of neuronal excitatory IT over inhibitory Vip cell types ($p < 5.53 \times 10^{-12}$) and low expression of non-neuronal types. **CTG 2**, dominated by psychiatric [56] diseases, exhibits more balanced pan-neuronal expression and is low in non-neuronal types. **CTG 3**, representing the non-neuronal enriched tumor-based diseases, has pronounced non-neuronal expression and captures ADG 1 diseases from the whole brain analysis. Finally, **CTG 4**, associated with the neurodegenerative diseases, has predominant enrichment in *Vip* inhibitory neurons over excitatory and specialized non-neuronal types. The major cell types (inhibitory, excitatory, non-neuronal) of **Fig 3** differentiate the major disease groups of **Fig 1**, and corroborate the module-based analysis of **Fig 2C** for these diseases. Color consistency in the top annotation bars of **Fig 3** show that the data clusters both at the subclass type level *Vip*, *Sst*, *Pvalb*, *IT*, *L6b*, and non-neuronal types. Furthermore, analysis of variance at fixed cell types (**Fig Q in S1 Text**) shows that the highest variation across diseases occurs for excitatory and non-neuronal types. Interestingly, **Fig 3** illustrates gradients of increasing expression in excitatory cell types from **CTG 1–4** (CTG 3–4, $p < 0.0623$; CTG 2–4, $p < 3.56 \times 10^{-9}$; CTG 1–4, $2.93 \times 10^{-18}$) in *IT*, *ET*, and *L6b* cell types across CTG with enrichment in language development, obsessive-compulsive disorders (OCD), and epilepsy. While inhibitory variation as a class is not significant across cell type groups, vasoactive intestinal peptide-expressing (*Vip*) interneurons show, by contrast, a decreasing gradient in

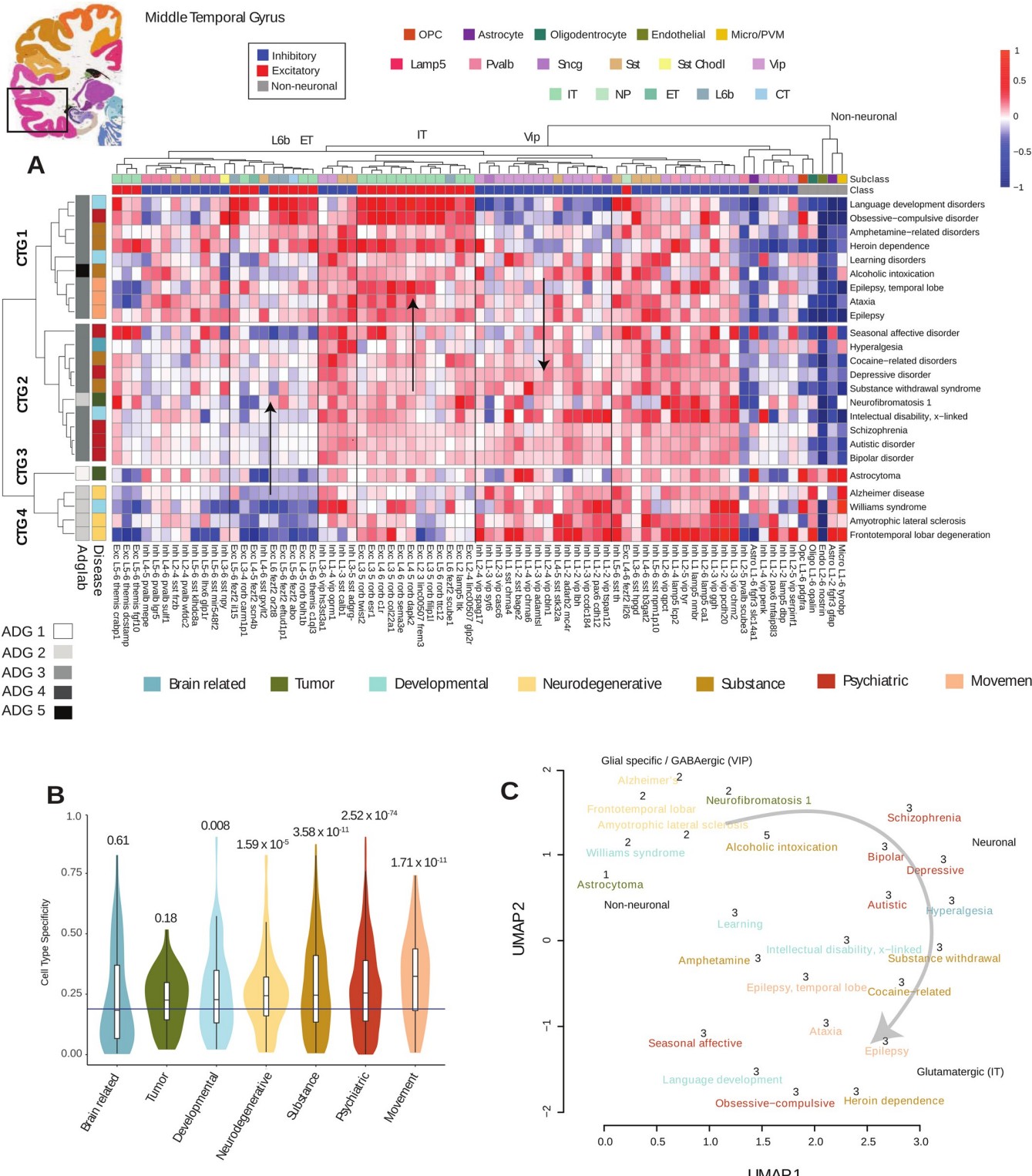

**Fig 3. Disease genes and cell types of middle temporal gyrus.** Coronal reference plate from the AHBA (http://human.brain-map.org) containing MTG region. (A) Mean cell type expression (CPM) of 24 cortex-related brain diseases (**Methods**) of 15,928 MTG nuclei over 75 cell types identified in [15]. Diseases and cell types are clustered and identify 4 cell type groups **CTG 1–4** based on cell type expression enrichment. Left annotation: **ADG** group membership determined by **Fig 1**, and GBD phenotypic classification. Top annotation: Major cell type classes (excitatory, inhibitory, non-neuronal) and subclass level inhibitory (*Lamp5, Sncg, Vip, Sst Chodl, Sst, Pvalb*), excitatory (*L2/3 IT, L4 IT, 5 IT, L6 IT, L6 IT Car3, L5 ET, L5/6 NP, L6 CT, L6b*), and non-neuronal (OPC,

Astrocyte, Oligodendrocyte, Endothelial, Micro-glial/perivascular macrophages). Color coding is by class (e.g., excitatory) and subclass types. Arrows indicate increasing and decreasing cell type expression gradients. (B) Cell type specificity τ measure pooled to phenotypic GBD categories shows psychiatric and movement classes as most cell type specific. Bar: mean specificity over all cells, p-values of each phenotype group show significance. (C) UMAP combining mesoscale and cell type disease relationships color coded by phenotype (**Methods**). Numbers show original ADG membership with primary cell type annotation and excitatory gradient. Underlying data for Fig 3 can be found in S1 and S6 Tables, and the data from S1 Data Disease Cell-type cluster level and correlation matrices. Raw data available at https://portal.brain-map.org/atlases-and-data/rnaseq under MTG SMART-seq(2018). Code available as a notebook at https://github.com/yasharz/human-brain-disease-transcriptomics. ADG, Anatomic Disease Group; AHBA, Allen Human Brain Atlas; GBD, Global Burden of Disease; MTG, middle temporal gyrus.

expression from CTG 1–4 (CTG 1–2, $4.09 \times 10^{-10}$; CTG 1–4, $8.26 \times 10^{-11}$; CTG 2–4, 0.0006). Here, pronounced enrichment of *Vip* interneurons, regulating feedback inhibition of pyramidal neurons [57], is seen in Alzheimer's disease [58], frontotemporal lobar degeneration, ALS [59], and Williams syndrome [60].

The structural (**Fig 1**) and cell type analysis (**Fig 3**) and their grouping by phenotypic classes is consistent, despite data being limited to nuclei from a single cortical area (**Fig R in S1 Text**). We combine the mesoscale and cell type approaches, averaging disease gene expression correlation matrices for 24 cortical diseases (**Methods**) and forming a consensus UMAP **Fig 3C** that graphically illustrates the transcriptomic landscape of major cortical expressing brain diseases, with key congruences and differences with phenotype association. The embedding in **Fig 3C** shows grouping by original **ADG**, colored by phenotype, with labeling of primary cell types, and the excitatory cell type gradient in cortical expression.

There is evidence in the literature consistent with a gradient in expression among these disease risk genes. Drugs of abuse have been shown to strongly alter neuronal excitability of layer 5 pyramidal cell types [61] and the largest transcriptomic change in epilepsy have been found to occur in distinct neuronal subtypes from the cell types L5-6_Fezf2 and GABAergic interneurons *Sst* amd *Pvalb*, consistent with higher expression in these CTG 1 diseases [62]. Further, the comorbidity of temporal lobe epilepsy with OCD [63] and with language development [64] is established. Genes associated with psychiatric disorders (CTG 2) are known to be widely expressed in the cortex [13], and GWAS studies in schizophrenia and depression show broad expression of susceptibility genes across neuronal cell types [65,66]. There is also increasing evidence that *Vip* expression is altered in numerous neurodegenerative disorders (CTG 4) [67] and the role of glial cells and their interactions with neurons is increasingly studied in neurodegenerative processes [68,69]. Co-expression relationships confirm these known associations linking diverse phenotypic disease groups.

## Excitatory cell type variation in psychiatric disease

The primary psychiatric diseases autism, bipolar disorder, and schizophrenia exhibit a largely similar expression profile (**Fig 3A**), but detailed variation is overshadowed by stronger variation in other disease groups, and by the large number of genes associated with these 3 diseases. These disorders with a heritability of at least 0.8, are among the most heritable psychiatric disorders and show a significant overlap in their risk gene pools [56]. We formed 3 matrices for the diseases autism, bipolar disorder, and schizophrenia, where each matrix measures covariation of cell type expression between MTG cell types (using genes unique to that disorder) and are independently thresholded for significance (**Methods**). Using these matrices, we investigate significant covarying cell types unique to autism, bipolar, and schizophrenia (*Aut*, *Bip*, and *Scz*), as well as those specific to pairs of diseases (*Aut-Bip*, *Aut-Scz*, and *Bip-Scz*) (**Fig 4A** and inset). Interestingly, excitatory variation dramatically exceeds inhibitory and non-neuronal variation for these diseases [70] accounting for 70.7% of significant cell type interactions. In particular, we find Aut-Scz (green) interactions with cell types of superficial layers

(*Linc00507 Glp2r*, *Linc00507 Frem3*, *Rorb Carm1p1*), *Bip-Scz* in intermediate layer types (*Rorb Filip1*, *Rorb C1r*), and a unique enrichment of bipolar risk gene expression in *Rorb C1r*. Remarkably, although the genes enriched in a given cell type differ between the 3 disorders (**Fig S in S1 Text**), specific neuronal circuits are shared between the diseases [71,72]. **Fig 4C** shows associated biological processes and pathways of the genes unique to *Aut*, *Bip*, *Scz* (g = 19, 20, 25) that pass the threshold in the interaction map of **Fig 4B** (**S7 Table**). The graph illustrates differential phenotypes, with genes uniquely associated with autism linked to brain development, schizophrenia-associated enriched genes implicated in dendritic outgrowth, and bipolar-associated genes linked to circadian rhythm [73]. The expressions of these unique genes have distinct profiles across the implicated cell types, with schizophrenia exhibiting pan-excitatory expression (**Fig S in S1 Text**). Cell type-specific interrogation of risk gene expression profiles provides insight into how polygenic risk might impact distinct types of neurons and neuronal circuits in psychiatric diseases while affecting overlapping pathways and processes.

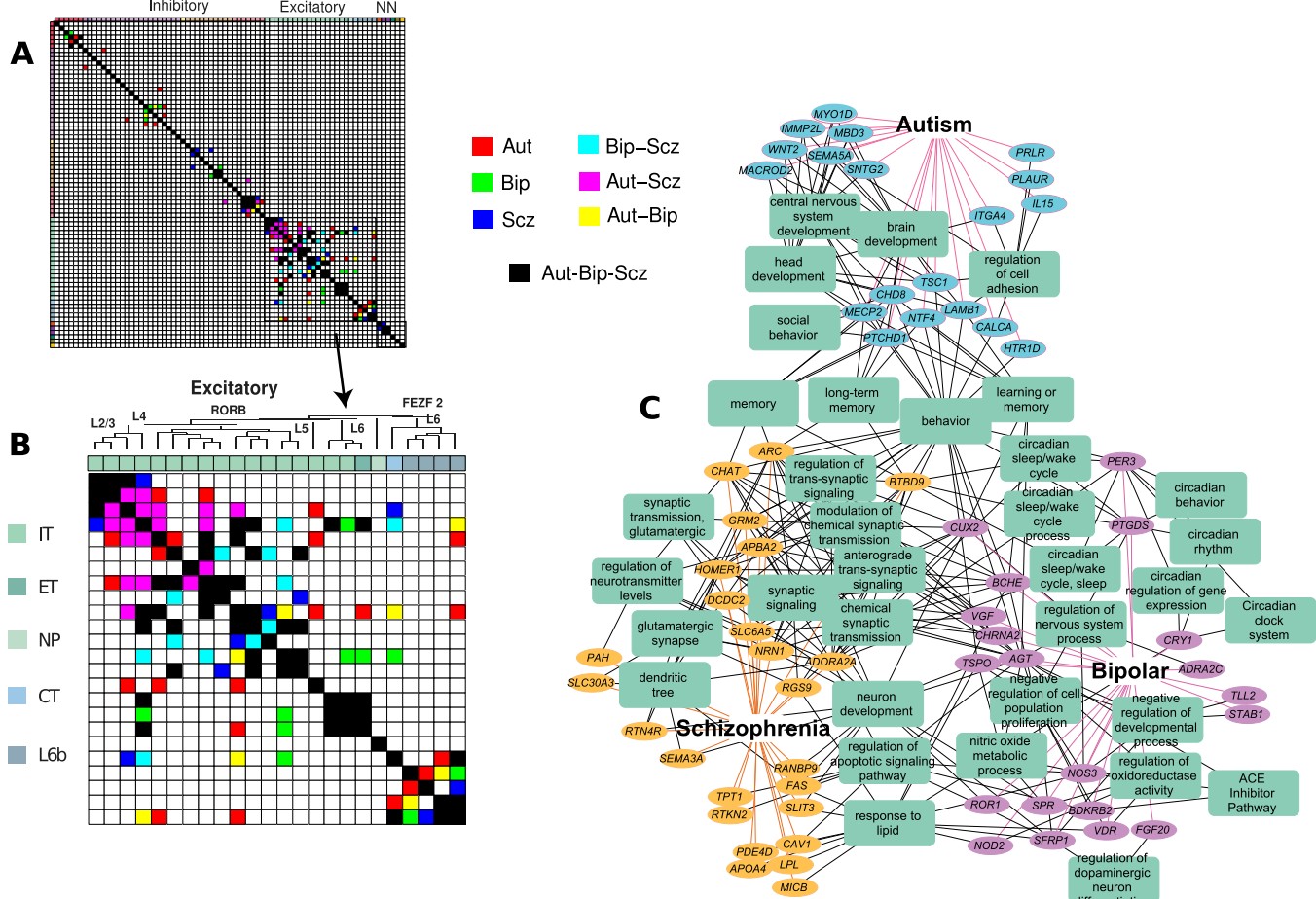

**Fig 4. Cell type profile of autism, bipolar, and schizophrenia in human MTG.** (A) Significant cell type-specific covariation of gene expression across MTG for 3 major psychiatric disorders (**Methods**). All 75 cell types from [15] with magnification of 24 excitatory types shown in (B), color coded by disease combinations. Autism (*Aut*, cyan), bipolar disorder (*Bip*, purple), and schizophrenia (*Scz*, yellow) show interactions unique to these diseases, *Aut-Bip* (blue), *Aut-Scz* (green), and *Bip-Scz* (red) unique to pairs, *Aut-Bip-Scz* (black) for all. Excitatory cell types (*IT*, *ET*,*NP*, *CT*, *L6b*) and dendrogram taxonomy from [15]. (C) Cell type-specific genes unique to excitatory interactions (*Aut*, *Bip*, *Scz*) from (B) and representative enriched biological processes and pathways. NN = non-neuronal. Underlying data for Fig 4 can be found in S1 and S7 Tables, and the data from S1 Data/Three_psychiatric_disorders. Raw data available at https://portal.brain-map.org/atlases-and-data/rnaseq under MTG SMART-seq(2018). Code available as a notebook at https://github.com/yasharz/human-brain-disease-transcriptomics. MTG, middle temporal gyrus.

## Brain diseases in mouse and human cell types

Single-cell profiling allows the alignment of cell type taxonomies between species, analogously to homology alignment of genomes between species. To examine conservation of disease-based cellular architecture between mouse and human, we used an alignment [15] of transcriptomic cell types from human MTG to 2 distinct mouse cortical areas: primary visual cortex (V1) and a premotor area, the anterior lateral motor (ALM) cortex. This homologous cell type taxonomy is based on expression covariation and the alignment demonstrates a largely conserved cellular architecture between cortical areas and species, identifying 20 interneuron, 12 excitatory, and 5 non-neuronal types (**Fig 5A**). We use this alignment to study species-specific cell type distribution over the 24 cortex disease groups both at resolution of broad cell type class ($N = 7$, e.g., excitatory), and subclasses ($N = 20$) where non-neuronal cell types are common between both levels of analysis.

To identify cell type differences in brain disorders between mouse and human cell types, we used expression-weighted cell type enrichment (EWCE) analysis [74]. Briefly, EWCE compares expression levels of a set of genes associated with a given disease to the genomic background with similar gene set size, determining significance through permutation analysis and excluding disease-related genes (**Methods**). EWCE evaluates all genes in a disease simultaneously, identifies the distribution of cell type expression for the group, and can be interpreted as characterizing the profile of active enriched cell types of a disease. The correlation of EWCE values aligned between mouse and human (**panel A of Fig T in S1 Text**, $\rho = 0.633$) is reflective of broadly conserved expression patterns [13] with minimally significant (K-S test: D = 0.0916, $p = 0.03$) difference in global EWCE distribution (**Fig 5B**). More remarkably, simultaneous clustering of EWCE mouse and human aligned cell types (**Fig 5C**, mouse (orange), human (blue)) shows a pairing of most diseases between species and indicates highly conserved cell type signatures at the subclass level. Remarkably, **Fig 5B** shows that the EWCE enrichment signature for ataxia, autistic disorder, epilepsy, bipolar disorder, ALS, Alzheimer's disease, and schizophrenia, and others are closer to the same disease across species than to any other disease signature within species. **Fig 5D** presents a similar co-clustering of normalized expression values for each disease in mouse and human. However, here the data clusters by species specific profiles while preserving many phenotypic GBD associations (left annotation). By homology mapping of cell types across mouse and human, we therefore find that mouse and human disease risk genes act in homologous cell types while having distinct species-specific expression (e.g., psychiatric diseases).

Cell type-specific enrichment by EWCE corroborates specificity of major cell types and subclasses in both mouse and human. **Panel B of Fig T in S1 Text** presents the significant EWCE $p$-values (after false discovery rate (FDR) correction) among mouse and human cell types, showing that psychiatric and substance abuse dominate the inhibitory (64%) and excitatory (70%) enrichments. While find no significant enrichments in either species for several diseases after correcting for multiple comparison including astrocytoma, neurofibromatosis 1, and frontotemporal lobar degeneration, the inhibitory subclasses *Lamp5*, *Sncg*, *Vip*, *Sst Chodl* show increased enrichment in both species (*Sst Chodl*, cocaine; *Sncg*, autistic, bipolar). Unique inhibitory enrichments are more common in mouse (*Vip*, autistic, bipolar, cocaine), while unique human enrichments are far more common in excitatory subclasses (*L6 IT Car3*, bipolar; *L2/3 IT*, *L5 ET*, depressive; *L6 CT*, learning disorders), and the only unique non-neuronal enrichment found is in human microglia/PVM for Alzheimer's disease ($p < 0.0012$).

## Discussion

We presented a brain-wide molecular characterization of common brain diseases from the perspective of neuroanatomic structure, aiming to describe how major transcriptomic

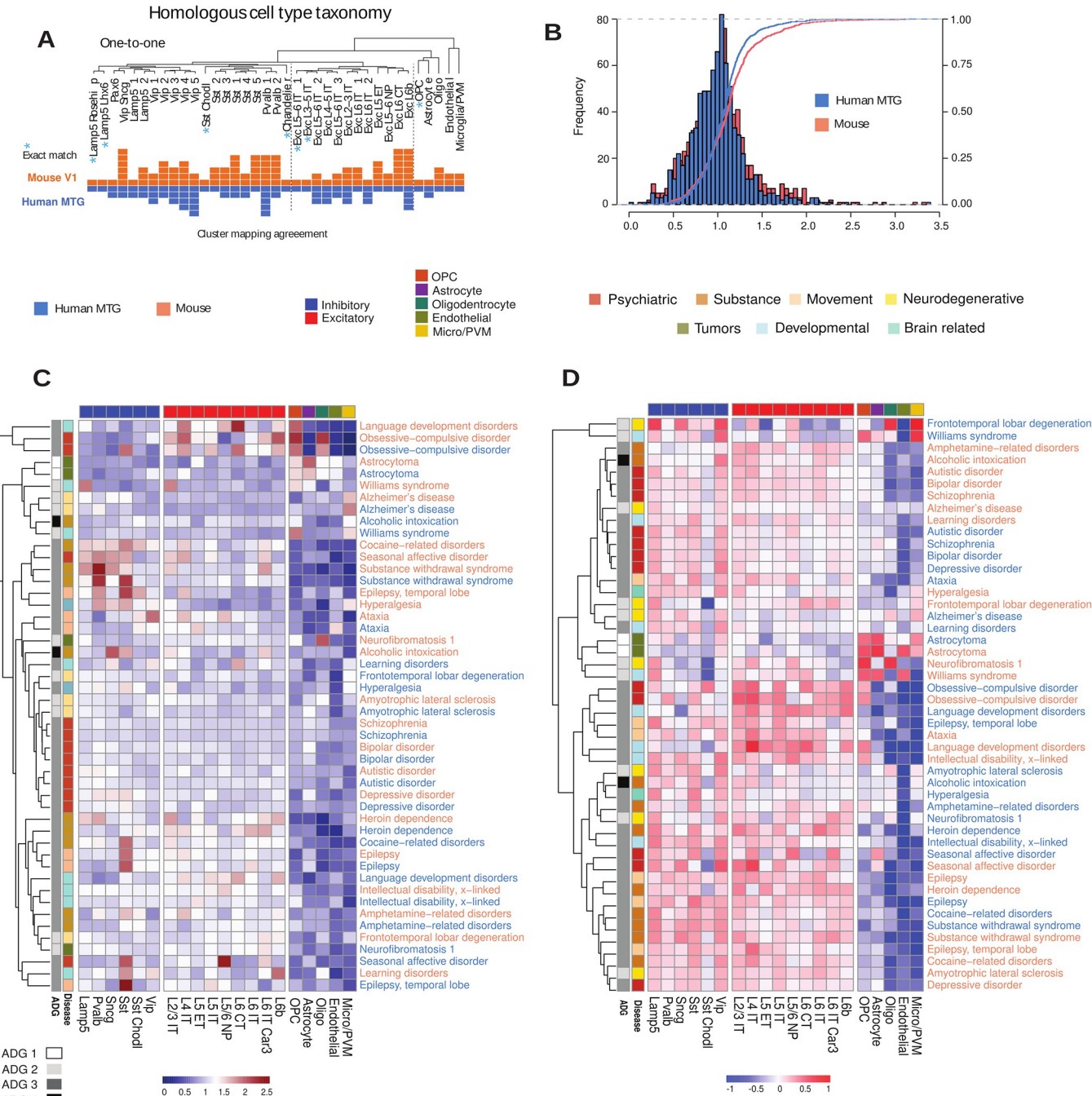

**Fig 5. Disease-based cell type expression in mouse and human.** (A) Alignment of transcriptomic cell types obtained in [15] of human MTG to 2 distinct mouse cortical areas, primary visual cortex (V1) and a premotor area, the ALM cortex, each square represents a mouse (orange) or human (blue) cell type cluster mapped to the homologous consensus cell type. (B) Histogram of mouse and human EWCE values [74] over subclass level of 20 aligned cell types. K-S goodness of fit test (**Methods**) shows that the distributions are marginally distinct (D = 0.091, p = 0.035). (C) Simultaneous clustering of mouse and human using EWCE disease signatures at subclass level 6 inhibitory, 9 excitatory, 5 non-neuronal (orange: mouse, blue: human) shows similarity of most diseases between species. (D) Similar clustering of mouse and human using average expression levels shows species-specific expression profiles while retaining GBD disease associations. Annotation top major cell classes, side disease GBD phenotype and ADG membership. Underlying data for Fig 5 can be found in S1 Table and the data from S1 Data (using EWCE_subclass as well as Cell_subclass expression files). Raw data available at https://portal.brain-map.org/atlases-and-data/rnaseq under MTG SMART-seq (2018). Code for EWCE available through https://github.com/NathanSkene/EWCE. Code available as a notebook at https://github.com/yasharz/human-brain-disease-transcriptomics. ALM, anterior lateral motor; EWCE, expression-weighted cell type enrichment; GBD, Global Burden of Disease; MTG, middle temporal gyrus.

relationships vary with common phenotypic classification. Precise phenotypic classification of diseases is challenging due to variations in manifestation, severity of symptoms, and comorbidities [4,73]. We used the Global Burden of Disease (GBD) study from the Institute for Health Metrics and Evaluation (www.healthdata.org) for high-level phenotypic categorization, as this work is a continuously updated, globally used, comprehensive, and a data-driven resource. While our approach cannot identify disease-specific gene expression changes precisely, we describe brain-wide transcriptomic architecture of genetic risk for major classes of brain diseases.

This study finds that diverse phenotypes and clinical presentations have shared anatomic expression patterns and may provide insight into disease mechanisms and frequency of comorbidity. Using anatomically mapped tissue sources and cell types, we observe that disease risk genes show convergent physiological-based expression patterns that associate diseases in expected and sometimes less expected ways. For example, language development disorders, OCD, and temporal lobe epilepsy are phenotypically diverse, yet all belong to **ADG 3,** and cell type analysis of **Fig 3A** indicates these diseases have a correlated cell type signature with strong IT excitatory subclass expression, and comorbidities identified in the literature. There is reproducible structure to these anatomic disease profiles illustrated through differential expression stability analysis and correspondence between mouse and human cell type profiles (**Fig 5C**). While the molecular basis of disease will ultimately reveal deeper associations which may lead to therapeutic options, our study is a step toward a biologically driven approach that uses transcriptomic and cell/pathway data to inform brain disorder classification.

For disease-associated genes, DisGeNet is one of the largest resources integrating human disease genes and variants from curated repositories and provides a standard approach to select genes for the study. Determining implicated genes in disease states presents considerable uncertainty, and any study is likely to miss important associations. Notably absent from our analysis are cerebrovascular diseases that account for the largest global burden of disability [4], and this limitation is due to relative under-sampling of rare vascular cell types in the Allen Human Brain Atlas. Also, the disease burden carried by each gene can vary significantly where the strength of the evidence supporting each gene varies, and the nature of the mutations causing each disease, or the mode of inheritance are essential to characterization. However, sources of variation are subtle, not well elucidated in the literature, and it is a major challenge of the translational studies to identify meaningful association and weights. The approach of DisGeNET prioritization relies on a statistical point of view where the affected brain structure, neural pathway, and cell type that can be identified is based on the normative expression profile of each gene. The utility of this assumption is potentially less meaningful when it comes to the effects of individual genes involved, and to address these issues, we conducted further analysis to evaluate the effect of gene importance as reflected in the literature. We used literature-based gene disease association weights provided by the DisGeNET dataset to allow for gene prioritization. The main disease categories show a very similar pattern across brain regions confirming the original classification to 85% agreement between class assignment of diseases.

The brain disorders included in this study have very different ages of onset and likely result from pathogenetic mechanisms active during different stages in lifespan. While the current study is performed with adult brain transcriptome data without considering developmental expression, genes that act in developmental period to cause pathology may continue to contribute to disease state in adulthood, and neurodevelopmental disorders have symptoms that are persistent across the life span. Although we do not claim to capture the developmental aspects of the disorders with our approach, it will still provide information about adult pathophysiology and it remains useful to elucidate these patterns in adults in comparison with other brain diseases. We have examined the presented set of diseases in the BrainSpan (https://www.

brainspan.org) data using donors from 60 days old to 39 years confirming known developmental trajectory of expression patterns and their convergence to adult patterning.

Brain-wide association of expression profiles may potentially implicate genes without previous association to a given disease, particularly when that profile is highly conserved between donors. The canonical transcriptional modules have been shown to be highly reproducible as default expression patterns in the adult [13]. The ability to associate genes through canonical expression patterns quantifies the global cell type distribution of expression related to disease risk genes. This has the potential of identifying new candidate risk genes not previously associated with disease risk. Similarly, brain-wide or regional expression datasets having divergent expression from normative in patients may provide clues to disease-specific alterations. We provide supplementary for the mapping of disease genes to modules and other closely correlated genes.

While previous work has shown conservation of neuronal enriched expression between the mouse and human [13,16], a recent novel alignment of mouse and human cell types in MTG now enabled a more specific analysis. For example, microglial involvement in Alzheimer's disease is well established, seen in **Fig 3** and found uniquely human enriched (**Fig 5B and panel B of Fig T in S1 Text**). Here, we show a striking conserved signature across subclass cell types for many diseases, and that the mouse appears to be evolutionarily sufficiently close to identify potentially relevant cell types, suggesting that we can leverage cross species cell type atlases to indicate disease risk gene patterning [75]. While homology alignment of cell types between mouse and human may provide insight into convergent mechanisms based on species-specific differences, further human data is needed to implicate disease genes with cell function.

The general correspondence of structural and cell type approaches even when restricted to a single cortical area (MTG) suggests a consensus organization and amplifies the value of cell type and tissue-based deconvolution methods, particularly when extrapolating these results to multiple brain regions. An intriguing finding is how diseases associated with pronounced cortical expression are organized along a gradient of excitatory cell types. This organization, also anti-correlated with an inhibitory gradient of specialized subclass interneurons, potentially provides insight into new methods for classifying cortical brain diseases. Cortical spatial gradients of gene expression were first observed in earlier tissue-based studies [14] and although originally attributed to sampling resolution have been now observed at cellular resolution [20,76]. With the increasing scale of single-cell studies, this may provide an important means of resolving cell type definitions and their relationship to disease.

A striking finding is the increased variability of excitatory cell types in psychiatric diseases (**Fig 4**) and certain species-specific expression differences in psychiatric and substance abuse diseases (**Fig 5B**). While there have been several lines of evidence that inhibitory cell types are impaired in the psychiatric disorders depression, bipolar disorder, and schizophrenia [77,78], results here indicate that excitatory pathways may be equally important. There are of course limitations to a cell type enrichment approach. Some diseases may involve gene pathways shared across cells rather than involvement of subsets of cell types or brain regions, and as others have found, cell type enrichment of disease genes does not necessarily match cell types with expression differences in disease versus control tissue [75,79]. Exploring the transcriptomic architecture of these disorders is a fully new field that has been underexplored and these findings support the transcriptomic hypothesis of vulnerability that in polygenic disorders, genes that are co-expressed in a certain brain region or cell type are much more likely to interact with each other than those that do not follow such a pattern [11,12].

Our results describe the structural and cellular transcriptomic landscape of common brain diseases in the adult brain providing an approach to characterizing the cellular basis of disorders as brain-wide cell type studies become available. The approach we present is flexible and

data driven and by following the steps in our accompanying *Jupyter* notebooks can be readily extended to multiple brain regions, with other diseases of interest and their associated genes, or updated with enriched or restricted gene sets. As cell type data is now being generated in multiple regions of the human brain through the Brain Initiative Cell Census Network (BICCN, www.biccn.org) and Brain Initiative Cell Atlas Network (BICAN), this work can be readily extended.

## Methods

### Disease genes

To obtain the gene disease associations, we used the DisGeNET database [21], a discovery platform with aggregated information from multiple sources including curated repositories, GWAS catalogs, animal models, and the scientific literature. DisGeNET provides one of the largest GDA collections. The data were obtained from the April 2019 update, the latest update related to the GDA at the time of analysis. An original list of 549 diseases from OMIM [13] with connection to the brain was intersected with the provided repository at DisGeNET. For each disease, the main variant was selected, and rare familial/genetic forms were not included in the analysis. For this study, we included genes with GDA reported at least in 1 confirmed curated (i.e., UNIPROT, CTD, ORPHANET, CLINGEN, GENOMICS ENGLAND, CGI, and PSYGENET) (for details, see https://www.disgenet.org/dbinfo). Since the goal of the study is to investigate the similarities and distinctions between brain-related disorders, disorders with less than 10 associated were excluded from the analysis. Finally, 15 disorders of peripheral nervous system or a second-level association to the brain (e.g., retinal degeneration) were removed. This procedure resulted in 40 brain disorders with their corresponding associated genes. Finally, for these 40 disorders, we performed a literature review of the current GWAS studies to add all the missing genes from the DisGeNET dataset. The 40 diseases include brain tumors, substance related, neurodevelopmental, neurodegenerative, movement, and psychiatric disorders (**Fig A in S1 Text**).

### Datasets

Anatomic-based gene expression data was extracted from 6 postmortem brains [14]. The extracted samples were divided into 132 regions based on the anatomical/histological extraction regions. These 132 regions were further pooled/aggregated into 104 regions including cortex (CTX, 8), hippocampus (HIP, 7), amygdala (AMG, 6), basal ganglia (BG, 12), epithalamus (ET, 3), thalamus (TH, 10), ventral thalamus (VT, 2), hypothalamus (HY, 16), mesencephalon (MES, 11), cerebellum (CB, 4), pons (P, 8), pontine nuclei (PN, 2), myelencephalon (MY, 12), ventricles (V, 1), and white matter (WM, 2) (**S3 Table**). The resulting gene by region matrix was averaged between subjects to produce 1 representative gene expression by region matrix and normalized across the brain regions. Cell type data is based on snRNA-seq from MTG largely from postmortem brains [15]. Nuclei were collected from 8 donor brains representing 15,928 nuclei passing quality control, including those from 10,708 excitatory neurons, 4,297 inhibitory neurons, and 923 non-neuronal cells. Cell type data from the mouse represents 23,822 single cells isolated from 2 cortical areas (VISp, ALM) from the C57GL/6J mouse [20].

### Uniqueness of disease transcriptomic profiles

Gene expression profiles across regions from each donor are correlated (Pearson correlation) to profiles from other donors and averaged to determine consistency of mapping to ADG and GBD groups and to identify exact disease associations between donors in **Fig 1**.

## Cell type specificity

Calculated based on the Tau-score defined in [55] and has previously been employed using the dataset [15]. Cell type specificity τ is defined as:

$$\tau = \frac{\sum_{1}^{N}(1 - x(i))}{(N - 1)}$$

where $x(i)$ is the gene expression level in each cell type for a given gene normalized by the maximum cell type expression of that gene, and the summation is over N cell-types in the analysis.

## Disease–disease similarity index

To calculate the similarity between each pair of disorders, we used the gene expression patterns across 104 brain structures. Distance metric between diseases is $1 - \rho$, where $\rho$ is Pearson correlation between structure or cell type profile. The procedure for disease similarity using cell type data used the gene expression pattern across the 75 cell types (instead of brain regions) in human cells extracted from MTG. For clustering in both cases, we used agglomerative hierarchical clustering with Ward linkage algorithm (Ward.2 in R hclust function, R version 3.6.3).

## Gene expression differential stability (DS)

Gene expression DS was calculated for each gene as the similarity of its expression pattern across 6 postmortem brains. For each pair of brains, the correlation of expression patterns across overlapping brain structures was calculated. The mean correlation over these 15 pairs was used as the DS for the given gene (for more details, see [13]).

## Disease-module association

Mapping gene expression for each gene to canonical modules, correlates the eigengene pattern from modules within each of 6 postmortem brains as explained in [14]. Correlation values are then normalized using Fisher r-to-z transform and averaged across brains. For each module, the gene associations were then standardized ($\mu = 0$, $\sigma = 1$). Finally, these values are averaged across genes associated with each disease to calculate the disease module association.

## Disease-related gene expression within cell types

We used EWCE analysis (https://bioconductor.riken.jp/packages/3.4/bioc/html/EWCE.html; [74]) to identify cell types showing enriched gene expression. EWCE compares the expression levels of the genes associated with a given disease to the background gene expression (all genes, excluding the disease-related genes) by performing permutation analysis and defining the probability for the observed expression level of the given gene set compared against a random set of genes with the same size. We used $N = 100{,}000$ as the permutation parameter and performed the analysis at 2 cell type category levels. The 2 levels included broad cell types ($N = 7$) and cell-subclasses ($N = 20$) with non-neuronal cell types common between the 2 levels of analysis. The 2 levels were selected due to the availability of the homologous cell types in mouse and human cell dataset. For each disease, we used FDR correction for multiple comparisons for disease-cell type associations for each disease.

### Cell type-specific interaction and functional enrichment

Gene expression covariation is computed as the absolute value of cosine distance similarity of cell type expression across MTG cell types. Matrices are computed for each of 3 psychiatric diseases using non-overlapping genes, and then independently thresholded to $1.5\sigma$. Entries are combined into a single matrix and are color coded if a given disease exceeds the threshold. Functional enrichment analysis to identify significantly enriched ($p$-value $<0.05$ FDR Benjamini and Hochberg) ontological terms and pathways for unique disease gene sets was done using the ToppFun application of the ToppGene Suite [80]. Representative enriched terms and genes were used to generate network visualization using Cytoscape application [81].

### Consensus representation

Consensus UMAP was constructed by averaging pairwise gene set correlation matrices for structural and cell type data and forming a 2D UMAP using R.

### Statistical analysis

All statistical analysis and visualization were conducted in R (www.r-project.org), a Jupyter notebook reproduces all analysis. To examine the differences in mean expression level between ADG groups, we performed ANOVA tests, followed by direct comparisons between ADG pairs using unpaired $t$ test. All results were corrected for multiple comparisons using Benjamini–Hochberg correction controlling the FDR. To examine the stability of the gene expression profiles, we repeated our analysis across 6 brains and searched for the matching pattern in other subjects for any given brain across ADG and GBD disease groups. Kolmogorov–Smirnoff test for goodness of fit is used in **Fig 5**.

### Supporting information

**S1 Text. Supporting Figures: Fig A in S1 Text. Classification and global burden of brain related diseases**. Major human brain diseases and classification according to the Global Burden of Disease (GBD) study [1,2] partitioned by 7 broad classes. The GBD study established the standard Disability Adjusted Life Years (DALY) metric to quantify disease burden defined as the years lost due to premature death plus years lived with disability. DALY scores are shown according to the 2019 study for several larger classes with error bars in white indicating minimum and maximum projected loss of life and healthy years. While cerebrovascular diseases including brain ischemia and infarction and related disorders dominate (global 2017 DALY 55.1 million, not shown), the combined toll of psychiatric disorders has nearly twice DALY (110 million). Neurodegenerative diseases account for less (38.2 million) primarily through older populations with Alzheimer's disease and related dementia (30.5 million) DALY. Color palette for these major GBD classes is used throughout the analysis. **Fig B in S1 Text. Neurological disorders and associated genes**. (A) Jaccard clustering based on relative percentage of shared genes (shown in gray scale color) between GBD classes for disease genes in this study. Inset numbers: number of genes in intersection, with diagonal total unique number to class. (B) Similar clustering of 40 neurological diseases and disorders. Top panel: fraction of genes uniquely associated with each disease. Color panel: membership GBD class for disease. Details of disease, gene sets, and metadata are given in **S1 Table**. Whereas the number of unique genes associated to GBD class psychiatric diseases (801) is 6 times larger than neurodegenerative diseases (132), a finer resolution does not reflect this bias with 110 genes (28.6%) unique to bipolar disorder, whereas 31 genes (30.3%) are unique to Parkinson's disease, 59 (88.0%) unique to hereditary spastic paraplegia. **Fig C in S1 Text. Biological process and**

**pathway ontology analysis** (www.toppgene.org) of genes uniquely associated with major GBD classes reflect common identifying annotations for these disease classes measured by FDR q-value. Color code in legend for GBD classes is used throughout the analysis. Specific associations of interest include well-known alterations in synapse structure and function (FDR q = $9.56 \times 10^{-50}$) [3], and abnormal levels of extracellular neurotransmitter concentrations [4] in several psychiatric and neurologic disorders (q = $1.25 \times 10^{-22}$). Major depressive disorder is one of the most important mental disorders associated with altered serotonergic activity [5], with less clear association in schizophrenia [6] and addiction [7]. Recent studies show that chronic type II diabetes mellitus (DM) is closely associated with neurodegeneration (q = $2.07 \times 10^{-5}$), especially AD [8]. The primary signaling pathway activated in insulin signaling is the phosphoinositide 3-kinase (PI3K)-protein kinase B (Akt) signaling stream, and defective IGF binding or IRS-1 signaling, as a result of insulin resistance, leads to cognitive decline in patients [9]. Hedgehog (Hh) is one of few signaling pathways that is frequently used during development for intercellular communication, important for organogenesis of almost all organs in mammals, as well as in regeneration and homeostasis. This includes the brain and spinal cord and mutations in the human *SHH* gene and genes that encode its downstream intracellular signaling pathway cause several clinical disorders, include holoprosencephaly [10]. Brain tumors and other cancers are strongly associated with defects in signal-transduction proteins., and cancers caused by certain viruses have contributed greatly to our understanding of signal-transduction proteins and pathways [11]. Chronic morphine-induced molecular adaptation of the cAMP cascade has been confirmed in many and has been widely related to opioid dependence and withdrawal [12]. These unique GBD class ontology annotations represent molecular function and pathways central to these major classes. **Fig D in S1 Text.** Transcriptome patterning of 40 brain diseases with clustering removing pairwise overlapping genes also identifies 5 anatomic groups. Most distinctive is the strong match of **ADG 1** and **ADG 2** demonstrating the identity and distinction of these groups. Removing common genes retains the association of the majority of **ADG 3** psychiatric, substance abuse, and movement diseases. The grouping of diseases in **ADG 5** is identically preserved in the clustering, overall indicating common structure with **Fig 1** and with pairs of diseases contained in the same ADG class with 67% agreement. **Fig E in S1 Text.** Clustering stability analysis for disorders with high gene count and overlap. To ensure that the co-clustering of psychiatric disorders is not the result of the high number of genes associated with these diseases as well as overlapping genes (see **Fig B in S1 Text**), we performed a clustering consistency analysis by sampling 200 genes from any disorder with more than 200 genes associated with it, and repeated the clustering analysis with the same *N* = 5 cluster size requirement. We then repeated this procedure 1,000 times and calculated the number of times each pair of disorders were co-clustered. The figure shows the frequency ratio of co-clustering across these 1,000 repeated analyses and indicates a stable cluster assignment. **Fig F in S1 Text. Reproducibility of ADG clustering.** A hold out analysis was conducted averaging the z-score normalized expression within each of the identified ADG groups identified in the full analysis of **Fig 1** with one of 6 brains data left out. On right annotation, 1 ADG 1 indicates that brain 1 data was removed and diseases in ADG groups averaged in the remaining 5 brains. Data is presented over 57 structures common to all 6 brains. Viewed as rows across structures, the reproducibility of expression patterning is seen to be highly consistent across hold out datasets with average correlation (ADG 1, ADG 2, ADG 3, ADG 4, and ADG 5) = (0.983, 0.971, 0.976, 0.988, and 0.977). Viewed as columns across structures the patterning has consistent differential expression across ADG groups. The annotation bar on top of the heatmap shows the maximum repeatable differential signature observed in each structure. The signature is exact (6) in all hold out brains for 27 structures and agree in all but one for 19 additional structures, only LA,

PRF, and Arc displaying variability. The expression signature itself is computed and compared as follows. For each structure and each hold out dataset the z-scored expression values are rank ordered giving a permutation of 1, 2, 3, 4, 5 from lowest to highest across the **ADG 1–5**. Each expression pattern is assigned a unique integer $n$ through unique prime factorization as $n = 2^{(1)}3^{(2)}5^{(3)}7^{(4)}11^{(5)}$ and these integers are tabulated to find the most occurring pattern across hold out brains. The maximum occurring signature 3–6 is shown in the annotation bar indicating similar conservation of signature to the hold out analysis, with 6 representing the exact relationship of ADG groups in all brains. **Fig G in S1 Text. Holdout analysis and ADG.** (Diagonal and upper) In each of 6 Allen Human Brain Atlas (AHBA) subjects, the mean disease transcription profile for each of 40 diseases across structures is computed and the most similar (Euclidean distance) disease in the remaining 5 subjects is identified. The upper diagonal matrix shows the distribution of identified diseases with key 0–6 indicating the number assignments to given disease. Thus, ataxia with score 6 has a transcriptomic profile more similar to ataxia for each brain than to any other disease in the remaining brains. Since the closest neighbor is an asymmetric definition, the average of the matrix and its transpose is presented. A majority 29/40 diseases are uniquely identified by majority voting. ADG groups 3, 4, and 5 have high identifiability across subjects while there is higher misclassification between ADG 1 and 2. Percent exact as in **Fig 1C** is ADG 1–5 (0.716, 0.537 0.644, 0.958, 0.875). Color bar shows Global Burden of Disease (GBD) groups. (Lower diagonal) A more stringent hold out analysis is conducted first eliminating common genes between the diseases as in **Fig 1** and by seeking the closest disease in transcriptome profile other than the given disease. Here, the distribution of disease mapping between brains is more variable having within ADG mapping **ADG 1–5** (0.361, 0.187, 0.970, 0.175, 0.008). **Fig H in S1 Text. Weighted gene clustering of brain disorders.** In order to evaluate the effect of gene importance as reflected in the literature, we used the literature-based gene disease association weights provided by the DisGeNET dataset. Each gene–disease association (GDA) has a score based on the following formula: **GDA-score = C + M + I + L**, where C is based on curated data sources, M is based on mouse and rat animal model reports, I is inferred GDAs from the Human Phenotype Ontology, and GDAs inferred from VDAs reported by Clinvar, the GWAS catalog and GWAS db, and finally, L is based on number of publications reporting the given GDA. More specifically, $C(N_1) = 0 + 0.3 \times (N_1 == 1) + 0.5 \times (N_1 == 2) + 0.6 \times (N_1 > 2)$, and $N_1$ is number of curated sources including CGI, CLINGEN, GENOMICS ENGLAND, CTD, PSYGENET, ORPHANET, and UNIPROT; $M(N_2) = 0 + 0.2 \times (N_2 > 0)$, $N_2$ is number of sources from Mouse and Rat from RGD, MGD, and CTD; $I(N_3) = 0 + 0.1 \times (N_3 > 0)$, $N_2$ is number of sources from HPO, CLINVAR, GWAS-CAT, and GWASDB; $L(N_4) = 0 + N_4 \times 0.01 \times (N_4 <= 9) + 0.1 \times (N_4 > 9)$, $N_4$ is the number of publications supporting a GDA in the sources LHGDN and BEFREE (see details in https://www.disgenet.org/dbinfo). Using the **GDA-score** for each gene disease association, we then calculated a weighted average expression representing the disease-related global gene expression pattern across brain regions that replaces the equally weighted gene expression average. Using this approach, we redid the main analysis for the AHBA dataset. The results show the new approach preserves the main disease categories going from tumor and neurodegenerative disorders toward psychiatric and motor disorders, with a very similar expression pattern across brain regions going from subcortical nuclei to cortical expression as observed in **Fig 1A**. Overall pairwise disease ADG membership agrees with the original clustering at 85%. **Fig I in S1 Text. Temporal evolution of average gene expression across 40 brain disorders.** The mean disease-related gene expression was calculated for each disease across brain regions for each time point using BrainSpan dataset (https://www.brainspan.org/) across developmental and adult years. Interestingly, tumor-based disorders expressing genes involved in regulation of cell population proliferation (see **Fig C in S1 Text**) have a biphasic early life and late

expression pattern, while developmental disorders show an early expression and drug abuse and psychiatric disorders show higher expression later, followed by a later stage expression in certain movement related and neurodegenerative disorders. We emphasize that one must be cautious to draw exact conclusions from these patterns since they are averaged across a multitude of genes and brain structures with heterogeneous gene expression patterns and this figure only shows the most dominant modes of expression across lifespan that survive in the averaging process. Based on proximity in the hierarchical clustering, the clustering preserves many of the adult associations based on proximity in the dendrogram. Annotation shows that GBD associations of diseases moderately agree. **Fig J in S1 Text. Pairwise comparison of ADG.** Pairwise B&H corrected (BH < 0.05) $t$ tests between **ADG** groups 1–5. Individual $t$ tests highlight the distinction in cortex expression between ADG 3 and other groups. The most significant structural ADG differences occur between **ADG 1–3** in cortex (frontal lobe (FL, $p < 2.71 \times 10^{-7}$)), short insular gyri (SIG $6.2 \times 10^{-9}$), long insular gyri (LIG, $5.57 \times 10^{-8}$), in amygdala, basolateral nucleus (BLA, $1.8 \times 10^{-9}$), basomedial nucleus (BMA, $4.49 \times 10^{-10}$), in cerebellar nuclei, globose nucleus (Glo, $1.18 \times 10^{-9}$), and myelencephalon, vestibular nuclei (8Ve, $2.34 \times 10^{-8}$). **ADG 2 and 3** are distinguished in hippocampus, (CA1, $2.18 \times 10^{-8}$), subiculum (S, $8.31 \times 10^{-8}$), in amygdala (AMG), amygdalo-hippocampal transition zone (ATZ $1.94 \times 10^{-10}$, BLA, $1.00 \times 10^{-10}$, BMA, $5.63 \times 10^{-10}$), and between **ADG 3 and 4** thalamus, anterior group of nuclei (DTA, $3.01 \times 10^{-7}$), lateral group of nuclei, dorsal division, (DTLv, $6.47 \times 10^{-9}$), and hypothalamus, posterior hypothalamic area (PHA, $1.21 \times 10^{-6}$). While there is not significant variation in the thalamus (TH, $p = 0.338$), myelencephalon (0.247), and cerebellum (CB, 0.966), differential telencephalic expression between psychiatric, substance abuse, and movement groups (**ADG 3**) and other ADGs is demonstrated by applying paired $t$ tests between groups. Here, **ADG 1** and **ADG 3** are distinguished through differences in frontal lobe (FL, $p < 2.71 \times 10^{-7}$), hippocampus, dentate gyrus (DG, $p < 3.46 \times 10^{-6}$), and amygdala, basomedial nucleus (BMA, $p < 4,49 \times 10^{-10}$). Finally, **ADG 4** and **5** differences are characterized by diencephalon expression: thalamus, anterior group of nuclei (DTA, $p < 3.01 \times 10^{-7}$), lateral group of nuclei, dorsal division (DTLv and hypothalamus, posterior hypothalamic area (PHA, $p < 1.21 \times 10^{-6}$)). **Fig K in S1 Text. Expression levels of brain and non-brain diseases.** (A) Expression levels of genes from Allen Human Brain Atlas (AHBA) classified as brain disease associated from this study (green), non-brain brain disease associated from OMIM study of [13] (gray) and remaining genes of AHBA not in these sets (red). Brain disease genes do not have significant expression differences from non-brain related genes, but both are different from non-disease associated genes with marginal significance. (B) Distribution of differential stability (DS) by major Global Burden of Disease classes. Horizontal mean $\rho = 0.521$ of 17,348 genes, with $p$-values shows significance (corrected for class size) of GBD mean differing from global mean. (C) Disease gene stability for 40 diseases sorted by median DS; colors are GBD classification. Minimum and maximum stable genes for each disease are shown. DS: differential stability. The set of high DS genes annotated (right) is substantially enriched for Gene Ontology biological processes and pathways compared to lower DS (left). **Fig L in S1 Text. Anatomic markers for DS genes.** For each of the 40 diseases, the highest and lowest differentially stable (DS) genes are selected. This results in 36 unique genes for low DS and 32 for high DS whose expression profiles are shown top (low DS) and bottom (high DS). High DS genes select for structural anatomic markers and cell types. This general expression consistency, less randomness, and reduced variation is seen for the expression profile of high DS genes. **Fig M in S1 Text. Disease-associated canonical expression modules.** Canonical module M1-M32 expression patterns are highly consistent across all 6 AHBA individuals, and patterns identified using any 5 brains could be found reproducibly in the sixth [13]. The modules range from structure-specific markers to complex co-expression patterns in the data, and several of the

modules are specific to the **ADG 1–5** groups. In addition to M1, M12 cited in the manuscript, M2 defines hippocampal expressing genes and M6 cortex-hippocampus co-expression; both are strongly represented by diseases in **ADG 3**. Representative genes and their correlation to the module eigengene are shown, PRKCA, STX1A is implicated in schizophrenia [14,15], ITGA4, MEF2C in autistic disorder [16,17]. M10 defines striatum expressing genes and is common among ADG 3 and 4 diseases. ADORA2A has been studied in amphetamine-related [18], depressive disorders and schizophrenia [19], and ANO3 in dystonia [20], Parkinson's disease, ALDH1A2 in Parkinsonian disorders [21] and schizophrenia [22], SEMA5A, autistic disorder [23]. Modules M24 and M25 are highly glial enriched and common in **ADG 1 and 2** diseases and effectively absent in ADG 3–5. FANCG has been studied in neurofibromatosis 1 [24], PPM1D in glioma [25], AIF1, Parkinson's disease [26], and TREM2 in Alzheimer's disease [27], amyotrophic lateral sclerosis [28]. **Fig N in S1 Text. ADG group comparison within canonical modules.** Corrected *t* tests between ADG groups for average disease correlation to the 32 canonical modules **M1-32.** Each set of data in the test consists of the correlation values in **Fig 2C** for those diseases in the corresponding ADG group at a fixed module. The tests are performed for all 6 pairs and each module independently. The -log10 Benjamini–Hochberg corrected values shown further validate the clustering of **Fig 1** and provide more insight into the cell patterning of ADG groups. **Fig O in S1 Text. Holdout analysis on canonical modules and ADG.** Comparison of holdout analysis for mean profile of **Fig 1** and based on canonical modules **Fig 2**. (A) Reproduction of holdout analysis for AHBA mean profile as in **S6 Fig** (upper diagonal.) In each of 6 Allen Human Brain Atlas (AHBA) subjects, the mean disease transcription profile across structures is computed and the most similar (Euclidean distance) disease in the remaining 5 subjects is identified. The matrix shows the distribution of identified diseases with key 0–6 indicating the number assignments to given disease. Perfect agreement in all subjects is a 6. (B) Similar analysis using canonical module assignments for 6 AHBA brains. Module-based assignment shows better definition of **ADG 1 and 2** and less variance in **ADG 3** with main psychiatric diseases, bipolar, schizophrenia, autistic disorder, and depression more closely identified. (C, D) Classification results by ADG and GBD categories. (E) Performance results for ADG and GBD comparing mean and module profiling. Mean is based on **Fig 1**, **Fig F in S1 Text** analysis; module based on canonical module assignments. ADG or GBD label indicates that the correct class was identified, Exact indicates that precise disease was identified. Mean ADG class is reduced 10% for modules but exact disease specification is improved 4%, while for GBD groupings there is both improvement of 4.5% across all classes and for 4% exact disease identification. **Fig P in S1 Text. Human MTG cellular data, expression level, specificity, and diseases.** (A) RNA-seq gene expression quantification with absolute expression levels estimated as counts per million (CPM) using exonic reads from [29]. (B) Cell type specificity was calculated based on the Tau-score ($\tau$) defined in [30]. This measure has previously been employed using the same dataset [29]. Distribution of $\tau$ for brain disease associated, non-brain disease, and unassociated genes. (C) Bar distribution plots for cell type specificity for 24 cortex expressing diseases, ordered by median specificity and colored by phenotypic GBD class. The correlation between the cell type-specific tau score and the mesoscale differential stability metric is 0.445. **Fig Q in S1 Text. Comparing cell type clusters (CTG).** Corrected paired *t* tests are used to compare significant expression differences between pairs of CTG groups, e.g., CTG 1 –CTG 2, at a fixed cell type. Overbar: ANOVA at each of 75 fixed cell types and clustered as in **Fig 3** over 3 CTG groups. The highest variability is seen among *IT* excitatory and non-neuronal cell types and at the subclass level GABAergic Vip cell types, consistent with the excitatory and inhibitory gradients of **Fig 3**. **Fig R in S1 Text.** (A) Clustering matrices for correlation between 24 cortically expressing diseases based on non-overlapping genes for both HBA and cell type MTG data. Data is shown for both matrices (upper

diagonal MTG, lower diagonal AHBA) with clustering based on MTG data of **Fig 3**. There is general structural correspondence of these matrices and overall disease–disease Pearson correlation between the matrices is $\rho = 0.615$. (B) For each of these 2D embeddings and each disease, the mean Euclidean distance from each disease to other diseases within the same GBD group is computed, as well as the mean distance to diseases not in that GBD group. The ratio of these quantities $GBD(d_i)$ is a measure of relative association of that disease with other diseases in the same GBD class. In symbols, as $GBD(d_i) = \mu_{d_j \in GBD}||d_i - d_j||/\mu_{d_j \notin GBD}||d_i - d_j||$. Diseases are then grouped by their GBD class showing general agreement between the approaches, except astrocytoma which is a significant outlier better classified using the mesoscale HBA data. Solid color: AHBA brain wide, dark gray: MTG cell type, light gray: consensus. **Fig S in S1 Text. Expression profiles of unique genes in autism, bipolar disorder, and schizophrenia.** Gene expression normalized for uniquely expressing genes in autism ($n = 19$), bipolar disorder ($n = 20$), and schizophrenia ($n = 25$) clustered by expression level over 24 excitatory cell types. The 3 diseases show distinct expression profiles across excitatory types with schizophrenia widely expressing most genes. **Fig T in S1 Text. Human and mouse EWCE distributions.** (A) Aligned transcriptomic taxonomy of cell types in human MTG to 2 distinct mouse cortical areas, primary visual cortex (V1), and a premotor area, the anterior lateral motor cortex (ALM) from [29] allows comparison of cell type enrichments between species. Scatterplot of disease-subclass EWCE values for mouse and human colored by **CTG 1–4.** Pie chart insets show percentages of CTG and GBD phenotypic classes of top 10% outliers from the regression line, representing most significant EWCE differences. Percentages (CTG 1, 0.363; CTG 2, 0.252; CTG 3, 0.220; CTG 4, 0.163). GBD Phenotype (Psychiatric, 0.137; Substance, 0.180; Movement, 0.125; Neurodegenerative 0.05; Brain tumors, 0.112; Developmental, 0.244; Brain Related, 0.150). (B) Significant species distinct EWCE based on FDR-correction of permutation based $p$-values by disease and cell type. **Fig 5C** of the main manuscript displays the EWCE values, whereas here, those values having significant $p$-values in either species are shown. Disease clustering is as in **Fig 3** with the same annotations and with color code (blue: human, orange: mouse, black: both species). Top barplot: number of cell type enrichments by species. (DOCX)

**S1 Table. Includes definitions, gene sets, and metadata identifying each disease.** First sheet table provides a general description of the disease with its traditional classification information and a link to each disorder's Medical Subject Heading (MeSH) webpage. Second sheet includes all the genes associated with each disease included in the current study. (XLSX)

**S2 Table. Includes the results for the functional enrichment analysis (https://toppgene. cchmc.org) of genes unique to each disorder, listing the major enriched biological processes and pathways and the corresponding statistical metrics for each entry.** (XLSX)

**S3 Table. Includes all the acronym, name, parent structure, and color code for each of the 104 structures from the Allen Human Brain Atlas (https://human.brain-map.org) included in the current study.** (XLSX)

**S4 Table. Includes the aggregated transcriptomic disease profile for each disorder.** Each sheet includes the aggregated gene expression for genes associated with a given disease across the brain structures listed in S3 Table. (XLSX)

**S5 Table. Includes the differential stability and associated canonical module, as defined in Hawrylycz and colleagues [13], for each gene included in the current study, sorted by the disease–gene pair name.**
(CSV)

**S6 Table. Includes 30 genes associated with brain disorders included in the current study that overlap with the 142 marker genes used to differentially distinguish the MTG cell types in Hodge and colleagues [15].** These genes form a highly differentially stable group, indicating strong cell type specificity, several uniquely associated with a disease.
(CSV)

**S7 Table. Includes a list of genes unique to autism, bipolar disorder, and schizophrenia, their corresponding enriched biological processes and pathways based on the functional enrichment analysis results (similar to the S2 Table) and select terms for their corresponding interactions network.**
(XLSX)

**S1 Data. Data accompanying our Jupyter notebook code to produce the main and supplementary figures in the manuscript, the data should be copied in a folder called input and the path should be added to the notebook file.**
(ZIP)

## Acknowledgments

The authors thank Christof Koch, Liane Ong, Stephen J. Smith, and Theo Vos for insightful and helpful discussions.

## Author Contributions

**Conceptualization:** Yashar Zeighami, Trygve E. Bakken, Thomas Nickl-Jockschat, Zeru Peterson, Anil G. Jegga, Jeremy A. Miller, Jay Schulkin, Alan C. Evans, Ed S. Lein, Michael Hawrylycz.

**Data curation:** Trygve E. Bakken, Anil G. Jegga, Michael Hawrylycz.

**Formal analysis:** Yashar Zeighami, Trygve E. Bakken, Thomas Nickl-Jockschat, Zeru Peterson, Anil G. Jegga, Michael Hawrylycz.

**Investigation:** Yashar Zeighami, Alan C. Evans, Ed S. Lein, Michael Hawrylycz.

**Methodology:** Yashar Zeighami, Trygve E. Bakken, Thomas Nickl-Jockschat, Zeru Peterson, Anil G. Jegga, Jeremy A. Miller, Jay Schulkin, Alan C. Evans, Ed S. Lein, Michael Hawrylycz.

**Resources:** Yashar Zeighami, Trygve E. Bakken, Alan C. Evans, Ed S. Lein.

**Software:** Yashar Zeighami.

**Supervision:** Michael Hawrylycz.

**Validation:** Yashar Zeighami, Michael Hawrylycz.

**Visualization:** Yashar Zeighami, Zeru Peterson, Anil G. Jegga, Michael Hawrylycz.

**Writing – original draft:** Yashar Zeighami, Trygve E. Bakken, Thomas Nickl-Jockschat, Zeru Peterson, Anil G. Jegga, Jeremy A. Miller, Michael Hawrylycz.

**Writing – review & editing:** Yashar Zeighami, Trygve E. Bakken, Thomas Nickl-Jockschat, Zeru Peterson, Anil G. Jegga, Jeremy A. Miller, Jay Schulkin, Alan C. Evans, Ed S. Lein, Michael Hawrylycz.

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
