## [Editor Report · Decision Letter 0]

11 Apr 2022

Dear Dr Zeighami, 

Thank you for submitting your manuscript entitled "Structural and Cellular Transcriptome Foundations of Human Brain Disease" for consideration as a Research Article by PLOS Biology.

Your manuscript has now been evaluated by the PLOS Biology editorial staff, as well as by an academic editor with relevant expertise, and I am writing to let you know that we would like to send your submission out for external peer review.

Once your full submission is complete, your paper will undergo a series of checks in preparation for peer review. Once your manuscript has passed the checks it will be sent out for review. To provide the metadata for your submission, please Login to Editorial Manager (https://www.editorialmanager.com/pbiology) within two working days, i.e. by Apr 13 2022 11:59PM.

If your manuscript has been previously reviewed at another journal, PLOS Biology is willing to work with those reviews in order to avoid re-starting the process. Submission of the previous reviews is entirely optional and our ability to use them effectively will depend on the willingness of the previous journal to confirm the content of the reports and share the reviewer identities. Please note that we reserve the right to invite additional reviewers if we consider that additional/independent reviewers are needed, although we aim to avoid this as far as possible. In our experience, working with previous reviews does save time. 

If you would like to send previous reviewer reports to us, please email me at kdickson@plos.org to let me know, including the name of the previous journal and the manuscript ID the study was given, as well as attaching a point-by-point response to reviewers that details how you have or plan to address the reviewers' concerns. 

Kind regards,

Kris

Kris Dickson

Neurosciences Senior Editor/Section Manager

PLOS Biology

kdickson@plos.org

---

## [Decision Letter · Decision Letter 1]

24 May 2022

Dear Yashar,

Thank you for your patience while your manuscript "Structural and Cellular Transcriptome Foundations of Human Brain Disease" was peer-reviewed at PLOS Biology. I apologize for the length of time that this took. Your manuscript has now been evaluated by the PLOS Biology editors, an Academic Editor with relevant expertise, and by several independent reviewers.

Given our and the reviewer interest in your study, we would be open to inviting a comprehensive revision of the study that thoroughly addresses all the reviewers' comments. As you will see in the reviewer reports, which can be found at the end of this email, the reviewers find the work potentially interesting. However, I will stress that the reviewers have also raised a substantial number of important concerns with the underlying assumptions that went into your gene selection and categorization processes that they feel impact how clearly the existing dataset can therefore inform on the underlying biological associations between these various and disparate disorders. Based on their specific comments and following discussion with the Academic Editor, it is clear that a substantial amount of work would be required to meet the criteria for publication in PLOS Biology. Given the extent of revision that would be needed, and that the outcome of these revisions on your core conclusions remains uncertain, we cannot make a decision about publication until we have a chance to assess your revised manuscript and your response to the reviewers' comments. If we felt that your revisions sufficiently addressed the reviewers' key concerns, we would then ask the reviewers to re-evaluate your work before making any further decision. 

We appreciate that the scale of the requested additional work is significant and you may well prefer to pursue publication of this work elsewhere. If you decide to continue consideration at PLOS Biology, we are willing to relax our standard revision time to allow you 6 months to revise your study. Please email us (plosbiology@plos.org) if you have any questions or concerns, or envision needing a (short) extension.

**IMPORTANT - SUBMITTING YOUR REVISION**

*Resubmission Checklist*

*Published Peer Review*

*PLOS Data Policy*

*Blot and Gel Data Policy*

Sincerely,

Kris

Kris Dickson, Ph.D. (she/her)

Neurosciences Senior Editor/Section Manager

PLOS Biology

kdickson@plos.org

REVIEWS:

Reviewer's Responses to Questions

PLOS authors have the option to publish the peer review history of their article (what does this mean?). If published, this will include your full peer review and any attached files.

Reviewer #1: No

Reviewer #2: Yes: Jose Davila-Velderrain

Reviewer #3: No

Reviewer #1: Zeighami et al. leveraged rich data resources (both neuroanatomical and single cell transcriptomes) from the Allen brain atlas to investigate the expression of disease-associated genes in adult human and mouse brain. The authors concluded that disease risk genes of different brain disorders exhibit different anatomic transcriptomic signatures. By analysis of single cell transcriptomes, they found cell type-dependent gradients that separate neurodegenerative, psychiatric, and substance abuse disorders.

The success of the authors' strategy hinges on gene selection for each of these complex and very different brain disorders. Each disease-associated gene carries equal weight in the analysis. This is potentially problematic because 1) the disease burden carried by each gene can vary significantly (some carry significant burden whereas others are risk factors), 2) the strength of the evidence supporting each gene also varies a great deal (some are convergently supported by multiple large cohort studies whereas others have conflicting data), 3) the nature of the mutations causing each disease (i.e. loss-of-function, gain-of-function, neomorphic, regulatory) or the mode of inheritance were also not considered.

For genes associated with autism spectrum disorder, some carry large effect in a nearly Mendelian way (e.g. CHD8), whereas others (e.g. MTHFR) carry weak and controversial association. As far as I can tell, these genes carries equal weight in the authors' analysis.

For genes associated with William syndrome, which results from copy number loss at 7q11, there is an additional issue. It is thought that only some of the genes within the CNV interval contribute to William syndrome phenotypes. It is therefore likely that some of the genes selected by the authors for analysis contribute to little or none of the disease (they merely fall within the CNV interval).

The nature of the disease-causing mutations needs to be considered. Genes can cause disease in many different ways. The authors' strategy may work well for loss-of-function mutations, but is likely not to work for gain-of-function, neomorphic, or regulatory mutations. The nature or direction of effect of mutations do not seem to have been considered; this is major caveat of the study.

The genetic architectures of the included brain disorders are very diverse. The current study design does not seem able to account for the contributions of common versus rare variants, modes of inheritance, levels of polygenicity, etc. 

It would be very helpful if the authors could provide at least some validation of some of their biological predictions. Even if empirical evidence is not possible, some orthogonal form of validation for at least a few of the biological predictions can add confidence to their analysis. For example, do these predictions align with what is known about disease etiology? Where do they disagree? Are there any ground truth data that the authors can benchmark their analyses against? 

It is important to note that the brain disorders included in this study have very different ages of onset and likely result from pathomechanisms during different times in the lifespan. The current study is performed with adult brain transcriptome data without taking into account developmental expression. This likely confounds the results. For example, genes that cause autism spectrum disorder likely affect prenatal development. The expression of these genes in the adult brain may be very different from fetal brain and may not be relevant to disease etiology. The absence of temporal expression analysis weakens the study. 

The significance of this work is dependent on the strength of the biological insights it provides into these brain disorders. Unfortunately, it is not clear that this work generated deep insights that can form the basis of future studies into disease mechanisms. 

Reviewer #2: Zeighami and collaborators present an integrative transcriptomic analysis of genes associated with 40 common brain diseases representative of 7 phenotypic classes. The authors report that diseases cluster in 5 groups determined by the similarity of expression patterns of their associated genes across anatomical structures of the adult human brain. These expression patterns are reproducible across subjects, generally discriminate among diseases, and only partially relate to phenotypic classes. Comparison with canonical gene expression modules from the Allen human brain atlas further supports distinctions between disease transcriptomic groups and suggest cell type associations underlying some of the differences. To further dissect these associations, the authors analyze expression patterns across cell types from the MTG for 24 diseases with preferential cortical expression. This analysis identified 4 disease groups based on cell type expression patterns and showed that gradients of expression across excitatory and inhibitory neuronal subtypes further distinguish disease groups. Finally, the authors show broad conservation and consistency of cell type enrichment patterns for disease associated genes in both human and mouse, with some exceptions suggestive of species-specific enrichment for a number of diseases and cell types.

Overall, their results and approach demonstrate that diseases can be compared and classified based on the neuroanatomic and cell type specific patterns of expression of their associated genes. The study provides an interesting example of how existing brain functional genomics data at different resolutions and in multiple species can be interrogated to better dissect and understand brain disease associations.

The approaches presented are interesting and timely, considering the increasing pace at which gene-disease associations and brain transcriptomic data in human and model species are being mapped. However, I have some concerns regarding the presentation of the results, some unclear methodologies, and the extent of biological interpretation. More clarity and additional biological interpretation complementing data description would largely benefit the study. 

See specific comments below:

The style and clarity of the text varies across the manuscript. The syntax and grammar of the first half of the manuscript should be revised. In particular, it is hard to follow the sections: "Introduction", "Brain disorders and associated genes", and "Structural transcriptomic profile of brain diseases". Likewise, the abstract should more closely summarize the data presented and highlight the main contributions. 

In section "Brain disorders and associated genes": It is not clear why and how the OMIM repository was used. Authors point to reference (14) to support the selection of 549 brain-related diseases to be intersected with the DisGeNET database. Reference 14 does not seem to be related to this. The information included in the associated methods section (Disease genes section) is largely a repetition of what is already included in the main text and does not clarify this issue. There are some inconsistencies between the numbers included in the main text and those included in the methods (e.g., "549 brain-related diseases" vs "an original list of 500 diseases"). Considering that all gene disease association data is coming from DisGeNET, please clarify why and how the OMIM repository was used. I would suggest clarifying and including most of these details only in the methods section.

Diseases are required to have at least 10 associated genes to be included in the study. However, several of the diseases included in Supplementary table 1 contain less than 10 genes. Are all the diseases in Supplementary table 1 included in the study or only a subset? An additional table sheet with descriptions for each data sheet would help clarify this and additional issues regarding the data presented in the tables.

In "the proportion of shared genes between diseases is known to be correlated with phenotypic similarity ( = 0.40, = 6.0 × 10−3)", it is not clear how these numbers were calculated and what they are referring to. How do you measure phenotypic similarity based on the data you have?

When listing gene distribution across GBD classes in the format (number, % unique to GBD class), the numbers shown are not percentages.

In the final disease/disorder list, what is "Dementia" referring to and how is it different from other common causes of dementia also included (e.g. Alzheimer's disease)?

What does structural transcriptomic profile mean?

In Figure 1A, an additional annotation column with the total number of genes for each disease (row) would help with data interpretation. Do diseases in ADG groups 4 and 5 tend to have less genes than diseases in other groups? If so, would that explain the lack of regularities seen in the other, larger ADG groups? An analysis demonstrating that differences in gene number do not play a major role in determining ADG patterns would improve this section.

In Figure 1A, it is not clear what uniqueness means. 

Are ADG expression patterns explainable by the degree of gene overlap within classes? It would be interesting to compare the degree of gene overlap (Jaccard index) between diseases of the same ADG group versus the overlap across ADG groups.

To what extent a small number of "influential" shared genes drives the associations? One way to address this could be by performing a reproducibility analysis similar to those presented in Supp Figues 5 and 6 but this time removing highly pleiotropic (genes) within each ADG group. This analysis would also complement the pairwise analysis presented later in Supp Figure 8. Alternatively, presenting earlier in the text a more extensive description of how the analysis in Supp Figure 8 addresses this problem -- perhaps with specific examples of particularly pleiotropic genes within classes -- would improve the section.

Figure legend explanations for panels C and D in Figure 1 are not very clear. Axes labels are missing. This analysis is very interesting, but the results are hard to follow as currently presented in the main text and figure. Are the authors trying to show that the anatomic pattern of a given disease in one subject tends to be similar to patterns in another subject for the same disease or disease of the same class -- and not to patterns of different diseases? 

This statement is not clear: "The ability to uniquely identify a disease from its anatomic signature indicates a finer transcriptomic patterning and is a bridge to cell type analysis".

Have the authors considered whether the fact that different diseases show different degrees of cross-subject anatomical profile similarity could relate to their underlying neurobiology? For example, given their developmental origin and high phenotypic heterogeneity, is it expected that psychiatric and developmental diseases show the least consistency? Some level of discussion of this would be interesting.

Brain disorders are classically defined based on observable neuropathological signatures (e.g., degenerative disorders) and/or behavioral symptoms (e.g. psychiatric disorders). There has been much discussion in the field regarding intrinsic limitations when trying to understand the neurobiology linking genes to brain disease phenotypes. Because multiple levels of organization are involved (molecular, cellular, circuit, behavioral, etc…), it is not clear whether certain accessible endophenotypic levels might be more appropriate than others to study disease. Have the authors considered interpreting/discussing some of their results in such a context? For example, some disease phenotypic classes show more consistent transcription patterns than others, and some diseases are more transcriptionally similar to diseases in other classes. Does this suggest that phenotype classes might not capture the relevant underlying neurobiology and need revision, or that molecular level endophenotypes are not equally informative across brain disease classes? Some level of discussion of these aspects would improve the representation of your results.

Regarding the use of existing canonical modules to aid interpretation, the following statement is very interesting, and perhaps could be expanded in the discussion section: "Brain wide association of expression module profiles may potentially implicate genes without previous association to a given disease, particularly when that profile is highly conserved between donors".

In "Averaging τ over sets of genes representing a given disease, we obtain a measure of cell type specificity of each disease within MTG (Suppl. Fig 14C)", figure reference seems incorrect.

The analysis in Figure 4 A and B is not very clear and the associated methods are very sparse. The axes are not labeled. Columns and rows seem to be cell types. What profiles are being used to compute covariation? What does cell type interaction mean in this context? How can single disease and disease-pair entries be defined based on this analysis? How do you go from this analysis to the genes in Figure 4B?

In Figure 5A, in the interspecies cellular taxonomy, it is not clear what the squares in the bottom represent. Is it the number of matches? Additional labels and more description would help.

In Figure 5B, it is not clear what scores are being shown. What type of scores EWCE analysis uses? Figure 5B shows positive numbers close to 1.0, while Figure 5C shows positive and negative numbers. Does the permutation analysis use z-scores to quantify enrichment? Are the values in Figure 5B -log(p-values)? If so, the fact that most values are close to 1.0 indicates that disease gene expression patterns are not cell type specific (enriched) similarly in human and mouse? Please clarify and add figure labels.

Figure 5D is not mentioned in the text. Perhaps the last paragraph should be referring to Figure 5D instead of Figure 5C.

The abstract mentions that comparisons with mice somehow indicate "where human data is needed to further refine our understanding of disease-associated genes". However, there is no data related to this point in the manuscript.

Add legends to numeric scales in all Figures -- including Supplementary.

Add labels to axes in all Figures -- including Supplementary.

When needed, include GBD legends in Figures to improve clarity.

I would suggest revising the title, in particular the word foundations does not provide any information.

Reviewer #3: This paper claims that 40 common brain diseases can be aggregated into 5 groups according to the anatomical expression of the genes associated with these brain diseases in the adult brain.

In this manuscript, gene sets for 40 brain diseases are collated from DisGenNET database, then the expression of these gene sets is examined in adult RNA-seq data from the Allen Human Brain Atlas across 104 structures and single-nucleus data for 75 cell types from the medial temporal gyrus. Gene expression data were averaged across gene sets for each disease and brain structure, then the gene expression data were clustered to define 5 Anatomic Disease Groups. Mean difference across groups and each structure were then calculated to quantify expression differences, so Mean of Mean analysis (statistician should evaluate the merits of this analysis). By investigating the co-expression modules initially reported for the Allen brain dataset, distinct modules are noted for brain regions. Brain disorder genes with high cerebral cortex expression were then examined for cell type enrichment in the MTG dataset. They also compare to mouse cell types.

The authors pose the following hypothesis: spatial and temporal co-expression of disease genes is indicative of a potential interaction between genes associated with brain diseases. 

Since the RNA datasets included in the analysis are derived from samples of adult brains, they provide no context for temporal relationship with gene expression. Similarly, co-expression is not a proxy for interaction, as RNA-protein are often not expressed at the same time or in the same cells (e.g., PMID: 35288716). 

Thus, the data do not address the stated hypothesis. Rather, the data reflect the following question: Do various brain disease aggregate based on the anatomical location of associated gene expression in the adult brain? The authors should address these limitations in their manuscript and edit their stated hypothesis, or include data (e.g. BrainSpain) that would allow for temporal analysis. In this context, there are many referrals to the patterning of the cortex in the results and discussion sections which seems to refer to the enrichment of gene expression patterns in the cortex which is different from enrichment in neuronal patterning, which is a developmental process. Thus the complete manuscript should be reviewed and edited for clarity.

Among the diseases included in the analyses, several are known to impact an overlapping set of brain structures - it would be helpful if the results were placed into this context of the known structures that are impacted. As already noted, many of the diseases investigated have known origins in early brain development, which is not addressed/discussed.

Cerebrovascular diseases, despite being the most burdensome brain diseases, were excluded from analysis due to the limited representation of relevant cell types in the datasets utilized. It would be helpful to provide a power calculation of the sample size needed for the RNA-seq datasets to be suitable analysis. Despite this stated limitation, the non-neuronal MTG cell types were included in analysis for those brain disease that were retained for analysis. More generally, it would be helpful to calculate the power for each analysis since the data and gene expression vary by region (number of genes expressed per region). Similarly, if the authors deconvolute the bulk cortex samples based on the MTG data, do they achieve similar results?

In the discussion, the authors should describe the novelty and implications of their results, which is not clearly described in the current version.

Overall, the current manuscript requires significant revision for clarity and context to be suitable for publication.

---

## [Decision Letter · Decision Letter 2]

16 Feb 2023

Dear Dr Zeighami,

Thank you for your (extreme) patience while we considered your revised manuscript "Anatomic and cellular transcriptome structure of human brain disease" for publication as a Research Article at PLOS Biology. This revised version of your manuscript has been evaluated by the PLOS Biology editors, the Academic Editor and the original reviewers.

Based on the reviews and our editorial discussions, we are likely to accept this manuscript for publication. While you will see that there is still a split in reviewer opinion on this work, we tended to agree with the more positively inclined reviewers that the strengths and limitations of the work are now more appropriately discussed. Given some continued concerns on these grounds however, we feel this work would be more appropriately published as a Methods and Resources article. When revising your work, please ensure upload your revision as a Methods and Resources article type. Please also ensure that you have addressed any remaining points raised by the reviewers with appropriate discussion, recognizing that our readership might have similar comments or questions. 

***In terms of reaching our broad readership, we'd also suggest a slight title change to: "A comparison of anatomic and cellular transcriptome structures across 40 human brain diseases"

***Please also provide a blurb which, if the paper is accepted, will be included in our weekly and monthly Electronic Table of Contents (eTOCs), sent out to readers of PLOS Biology. This blurb may also be used to promote your article on social media. The blurb should be about 30-40 words long and is subject to editorial changes. It should, without exaggeration, entice people to read your manuscript, should not be redundant with the title and should not contain acronyms or abbreviations. For examples, view our author guidelines: https://journals.plos.org/plosbiology/s/revising-your-manuscript#loc-blurb

***Please also make sure to address the data and other policy-related requests at the bottom of this email. IMPORTANT - failure to fully and completely address these points will delay further handling of your work at PLOS Biology.

We expect to receive your revised manuscript within two weeks. 

*Published Peer Review History*

*Press*

Please do not hesitate to contact me should you have any questions. Please also accept my apologies for the delays in getting back to you. Our Academic Editor was traveling and it took some time for us to discuss the work and the varied reviewer feedback amongst all of us.

Sincerely,

Kris

Kris Dickson, Ph.D., (she/her)

Neurosciences Senior Editor/Section Manager,

kdickson@plos.org,

PLOS Biology

DATA POLICY:

Note that we do not require all raw data. Rather, we ask that all individual quantitative observations that underlie the data summarized in the figures and results of your paper be made available. We appreciate that the raw data is available online. We ask that you also clearly indicate, for all your main and supplemental figures, what data was used to create each of the figure panels in the current study. 

***To do so, please direct readers, within each figure legend, to the correct supplementary table using a statement "The underlying data supporting Fig X, panel Y can be found in file Z.". 

Our aim is to ensure that our readers can easily reproduce and reanalyze the data presented in your study for all of your figures:

Fig 1A-D; Fig2A-C; Fig3A-C; Fig4A-C; Fig5A-D

Supplemental: Fig 1; Fig2A-B; Fig3; Fig4; Fig5; Fig6; Fig7; Fig8; Fig9; Fig10; Fig11A-C; Fig12; Fig13; Fig14; Fig15A-D; Fig16A-C; Fig17; Fi18A-B; Fig19; Fig20

DATA NOT SHOWN?

- Please note that per journal policy, we do not allow the mention of "data not shown", "personal communication", "manuscript in preparation" or other references to data that is not publicly available or contained within this manuscript. Please carefully check your submission for any such statements and either remove mention of these data or provide figures presenting the results and the data underlying the figure(s).

Reviewer remarks:

Do you want your identity to be public for this peer review?

Reviewer #1: No

Reviewer #2: Yes: Jose Davila-Velderrain

Reviewer #3: No

Reviewer #1: In this revised manuscript, Zeighami et al acknowledged many of the weaknesses pointed out by the reviewers in the last round of reviews. The revisions however were largely textual; the limitations, although now formally acknowledged, remain unresolved and still impact the analyses and the significance of the overall study.

Previously, I pointed out that the success of the authors' strategy hinges on gene selection for each of these complex and very different brain disorders. The problems are 1) the disease burden carried by each gene can vary significantly (some carry significant burden whereas others are risk factors), 2) the strength of the evidence supporting each gene also varies a great deal (some are convergently supported by multiple large cohort studies whereas others have conflicting data), 3) the nature of the mutations causing each disease (i.e. loss-of-function, gain-of-function, neomorphic, regulatory) or the mode of inheritance were also not considered. In the revised manuscript, the authors used the gene-disease association (GDA) score from DisGeNET, which takes into account the number and type of sources, and the number of publications supporting gene-disease association and presented the data in Suppl. Fig. 8. Although this analysis addressed point 2 above (on the strength of the evidence), point 1 (on disease burden/effect size) and point 3 (on the type of mutation or mode of inheritance) remain important issues that have not been resolved, limiting the strength of the analyses and the significance of the study as a whole. 

In the first round of reviews, another reviewer and I each raised the point that the brain disorders included in this study have very different ages of onset and likely result from pathomechanisms during different times in the lifespan. The use of adult brain transcriptome data without taking into account developmental expression likely confounds the results. The authors responded that they "observe that even genes that likely act mostly in development to cause pathology may continue to contribute to disease state in adulthood since those genes are still expressed, and neurodevelopmental disorders have symptoms that are persistent across the life span." What is the evidence supporting this statement? Genes that are required for brain development can be expressed in adult but play a much different or less functional role. They can also shift in their expression patterns from development to adulthood. Persistent adult symptoms from neurodevelopmental disorders can originate from altered developmental processes that have a lasting impact through life that have nothing to do with the adult expression of the gene. I do not understand the logic of the authors' argument here.

Furthermore, the authors "have also now examined the presented set of diseases in the BrainSpan (https://www.brainspan.org) data using donors from 60 days old to 39 years. The results highlight the expected temporal patterning and onset of expression in the diseases, while many of the adult associations presented in Figure 1 remain. We have placed this result in a Suppl. Fig. 9 and comment on these issues in the main text." I do not see what in Suppl. Fig. 9 supports the assertion that "many of the adult associations presented in Figure 1 remain"; 27/40 disorders now fall under a single large group; the other groups are a mix of diseases from previously disparate groups. Importantly, disorders with very different ages of onset, for example Autism and Alzheimer, fall under the same group. There are clear discrepancies compared to the original analysis.

An important issue that I previously brought up is that the significance of this work is dependent on the strength of the biological insights it provides into these brain disorders. Unfortunately, it is not clear that this work generated deep insights that can form the basis of future studies into disease mechanisms. They authors, "of course agree with the reviewer that a study at this resolution of analysis will not be expected to yield profound results about individual diseases." If this is the case, then what is the substantive contribution of this study overall?

In the present form, this work represents a set of predicted relationships between disorders that have not been orthogonally validated. In addition to the validity of the predictions, the real world utility of the work is also in question. The authors state that they "believe the work has value in generating hypotheses that may be followed up through experimental and computational approaches in further studies." However, I did not find where the authors described concrete, testable hypotheses generated based on their analyses. The authors argue that their study "is a step toward a biologically driven approach that uses transcriptomic and cell/pathway data to inform brain disorder classification." However, it does not seem to inform the clinical classification, diagnosis, or treatment strategies of these disorders. The real world impact of the work is therefore not clear. A major take home seems to be that "we observe that disease risk genes show convergent physiological based expression patterns that associate diseases in expected and sometimes less expected ways." This speaks neither to the validity nor the utility of the predictions. 

Reviewer #2: The authors have addressed all my previous comments and suggestions. The updated manuscript is much improved.

As final comments:

I suggest further revising the abstract to more clearly delineate what constitutes background knowledge, what are the analytical contributions of the current study, and what are the major findings. 

I suggest adding a description sheet to each Supplementary table to clearly describe the different data and features being presented.

Reviewer #3: The authors have thoughtfully and appropriately responded to the reviewers' comments. The substantial revisions in the current manuscript have clarified the results and interpretations and include important limitations.

---

## [Editor Report · Decision Letter 3]

2 Mar 2023

Dear Yashar,

Thank you for the submission of your revised Methods and Resources article "A comparison of anatomic and cellular transcriptome structures across 40 human brain diseases" for publication in PLOS Biology. On behalf of my colleagues and the Academic Editor, Nicole Soranzo, I'm pleased to say that we can in principle accept your manuscript for publication, provided you address any remaining formatting and reporting issues. These will be detailed in an email you should receive within 2-3 business days from our colleagues in the journal operations team; no action is required from you until then. Please note that we will not be able to formally accept your manuscript and schedule it for publication until you have completed any requested changes.

IMPORTANT: I note that your code is deposited in Github. Because this can be changed or deleted at any point, please can you generate a permanent DOI'd copy of this (e.g. in Zenodo, etc.), and include the relevant URL in your manuscript? I've left a note with one of my colleagues to include this request alongside their formatting requests.

Sincerely, 

Roli

Senior Editor

PLOS Biology

rroberts@plos.org